# Forest fires and climate-induced tree range shifts in the western US

Avery P. Hill 🆔 [1✉] & Christopher B. Field 🆔 [1,2]

Due to climate change, plant populations experience environmental conditions to which they are not adapted. Our understanding of the next century's vegetation geography depends on the distance, direction, and rate at which plant distributions shift in response to a changing climate. In this study we test the sensitivity of tree range shifts (measured as the difference between seedling and mature tree ranges in climate space) to wildfire occurrence, using 74,069 Forest Inventory Analysis plots across nine states in the western United States. Wildfire significantly increased the seedling-only range displacement for 2 of the 8 tree species in which seedling-only plots were displaced from tree-plus-seedling plots in the same direction with and without recent fire. The direction of climatic displacement was consistent with that expected for warmer and drier conditions. The greater seedling-only range displacement observed across burned plots suggests that fire can accelerate climate-related range shifts and that fire and fire management will play a role in the rate of vegetation redistribution in response to climate change.

[1] Department of Biology, Stanford University, Stanford, CA, USA. [2] Woods Institute for the Environment, Stanford University, Stanford, CA, USA. ✉email: aph82@stanford.edu

At the broadest scale, plant biogeography is largely determined by climate[1–3]. A panoply of research has supported the expectation that plant distributions should change in response to climate changes[4–8]. Range shifts over the last century have been documented around the globe, particularly in mountainous areas[9], with mounting evidence of changing distributions in flatter landscapes as well[8,10,11]. The required rates of plant range shifts to keep up with the velocity of climate change are large, relative to observed range shift rates[12]. Though these rates are highly variable among taxa, a global synthesis found that plant distributions have been shifting to higher elevations at an average rate of 1.5 m/y[8]. Of the plants that do track climate change—many do not[10,13] range shift rates are often too slow for species to keep pace with the contemporary velocity of climate change[14–16]. Furthermore, plants must contend with other anthropogenic factors such as land-use change (e.g., logging, urbanization) or invasive species. Our ability to understand and predict the distribution of vegetation in the Anthropocene is important for resource management, conservation, and other efforts to secure a sustainable future. The interspecific variability and context specificity of range shift rates make modeling range shift difficult. Research into the causes and ecological implications of variable range shift rates is ongoing and incomplete.

Range edges are dynamic over time, expanding or contracting as the relationship between a population and its environment changes, because of changes in (1) the environment (e.g., warmer temperatures, increased precipitation seasonality, etc.) and/or (2) the population (e.g., phenotypic changes in temperature tolerance, water-use efficiency, etc.). At time scales (1 to 2 generations) where evolution is unlikely to have made populations tolerant to observed climate changes[17], we can expect range expansions where environmental constraints that limit species ranges are lessened/removed. Features that affect the rate at which these range edges shift include species characteristics (morphology, dispersal mechanisms, etc.), landscape characteristics (suitable habitat availability, disturbance regimes, geographic barriers), and biotic interactions (interspecific competition, predation, mutualism)[18]. Of these, effects of biotic interactions may be the most difficult to understand and predict not for lack of effort[19] but in part due to the specificity, conditionality, and complexity of the relationship between species interactions and local population establishment and growth. While biotic interactions have both inhibited and facilitated range shifts during historical deglaciation events[20,21], empirical evidence of their role in contemporary range shifts is sparse.

Competition is known to affect range limits[22,23]. A hypothesis that the leading edge of migrating populations may be slowed by competition with pre-existing vegetation[18,24] is supported by some vegetation models[25–28]. However, empirical evidence of competition slowing leading-edge range shift in contemporary systems under climate change is scant[24]. Though some recent experimental work suggests that interspecific competition is more significant at the trailing-edge than the leading-edge in montane plant communities[29,30], a continental-scale and/or observational investigation into this phenomenon has not yet been conducted.

Wildfire, because it reduces vegetation cover and therefore reduces some aspects of plant–plant competition, provides an entry point for exploring the hypothesis that removing competitors can accelerate climate-related range expansion. Of course, the effects of fire on community assembly certainly go beyond reducing plant–plant competition, and other aspects of fire ecology could influence climate-related range shifts. Recent theory and modeling indicate that disturbance has the potential to facilitate range expansion[25,31–33], specifically forest fires[28,34], by reducing competitor population size and opening niche-space for colonization. Recent observations document that fire can drive range contractions across western North America[35–38], but evidence of range expansion facilitation is less concrete.

At the local scale, wildfire typically facilitates shifts in species composition along trajectories of ecological succession[39,40], where species that become abundant shortly after fire are eventually replaced. Post-fire succession can be relatively predictable and specific to a particular ecosystem type, often returning an ecosystem to approximately the pre-fire species assemblage. In contrast, fire can also facilitate state-shifts where the post-fire trajectory of vegetation succession leads to a different suite of species[3,36]. This kind of state shift has occurred within the boreal forests of Alaska, where fire decreased ecosystem resilience, i.e., the ability of an ecosystem to return to the pre-existing species composition after disturbance. The impact of fire on species regeneration (e.g., by altering seedbank composition) was cited as an important driver of the observed state shifts[41,42]. Fire and climate change, in concert, can affect tree regeneration as well. In one study on *Pinus ponderosa* and *Pseudotsuga menziesii* forests, warmer temperatures made conditions unsuitable for conifer seedlings that emerged after fire[43]. Ongoing rapid changes in fire and climatic regimes have the potential to drive widespread vegetation change across the western United States[36,43–45].

In this study we used observational evidence from the USDA Forest Inventory and Analysis (FIA) to test the hypothesis that forest fires can facilitate range expansions in tree species responding to contemporary climate change. We used all FIA plots (74,069 total) within the Northwestern Forested Mountains and Marine West Coast Forest ecoregions of the continental U.S. as a natural experiment, with historic wildfires as the treatment and a proxy for tree range shift rate as the response. We analyzed the difference in the climatic niches of mature and juvenile trees, much like Dobrowski et al.[46], and used the resulting Seedling Only Range Displacement (SORD) values to approximate range shifts in geographic space. We defined the source population as composed of plots with both trees and seedlings present and the potential leading-edge as composed of plots with only seedlings present (if these seedlings reached reproductive maturity they may have eventually become a true leading edge). In effect, we calculated range shift velocity using a combination of climate-space for geographic-space substitutions and age-for-time substitutions, focusing on the comparative analysis of plots with and without recent fire. Here, we show that the potential range shift rates of two of eight tree species are significantly faster in plots with a recent fire.

## Results

**Species selection.** Of the 110 tree species within the study area (Fig. 1), 26 were sufficiently abundant to meet the minimum plot-level sample size for both burned and unburned plots and for both seedling-only and tree-plus-seedling. We filtered out 14 species due to inconsistencies in the direction of SORDs between life stages, based on the angle between SORD vectors (Fig. 2), and an additional 4 due to directional inconsistency in SORDs between burned and unburned plots (Supplementary Table 1 includes more information on the species filtered out at each stage). The comparison of SORDs between burned and unburned samples is useful only for species where the direction of potential range shifts is consistent across populations. These eight remaining species are described in Table 1.

**Evidence of SORD in unburned plots.** Our comparative analysis of burned and unburned SORDs is meaningful only for species that demonstrate potential range shifts (SORDs) in the unburned (control) plots. The unburned seedling-only populations of all eight species grew in significantly different climates than

unburned tree-plus-seedling populations by two metrics. Schoener's D is a metric of climatic niche equivalency, where values closer to 1 indicate more niche similarity, and values closer to 0 indicate less. In unburned plots, the overlap between the climatic niche of the seedling-only and the tree-plus-seedling populations yield Schoener's D values from 0.85 to 0.64 (Table 2). All values of Schoener's D indicate less niche equivalency than would be expected at random ($p < 0.05$), leading us to reject the niche equivalency test's null hypothesis that the climatic niches of the seedling-only populations are equivalent to the niches of the tree-plus-seedling populations.

The unburned SORDs derived from Schoener's D of the eight species are corroborated by the 2nd metric, Euclidean centroid distance, which was calculated based on the 4-dimensional centroids of Mean Temperature of the Warmest Month (MTWM), Mean Temperature of the Coldest Month (MTCM), Mean Summer Precipitation (MSP), and Mean Winter Precipitation (MWP). All unburned centroid distances are greater than 0.19 (Table 2) and significantly greater than 0 (Hotelling's $T^2$ test[47], $p < 0.05$). Although the centroid distances were calculated

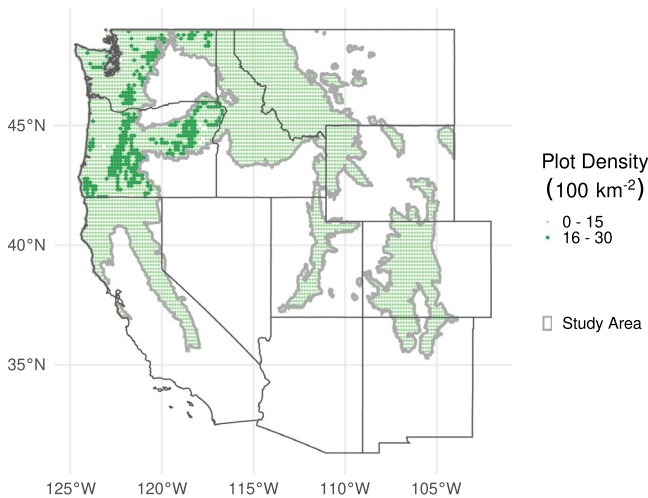

**Fig. 1 Map of the study area.** The Northwestern Forested Mountains and Marine West Coast Forest ecoregions (thin gray outline) of the northwestern continental United States determined the extent of the FIA plots (green points) that were sourced. Larger and darker points represent areas that were sampled at a greater geographic density. Source data are provided as a Source Data file.

in 4-dimensional climate space, PCA is useful for visualizing the differential clustering of component scores and the corresponding centroids (the average component score coordinates). Figure 3 and Supplementary Fig. 1 illustrate how, even though the climatic niches of unburned seedling-only and tree-plus-seedling plots can be broadly overlapping, the centroids of the 2 groups can still differ.

**Comparing SORDs between burned and unburned plots.** Although we use two metrics to characterize SORDs, Schoener's D lacks directionality, making it difficult to intuitively compare the SORDs of burned and unburned populations. For this reason, we use the difference between burned and unburned centroid distances ($CD_B - CD_U$; where B = burned, U = unburned) as our primary method of comparison. $CD_B$ is greater than $CD_U$ for both *Pseudotsuga menziesii* and *Quercus chrysolepis* ($p < 0.05$). The centroid distances of the other six species did not significantly differ between burned and unburned samples. The average trend across all eight species (with mixed $p$-values) is that $CD_B$ is 89% greater than $CD_U$ (Table 2).

When the SORDs are broken down into their constituent climate variables, five species—*Chrysolepis chrysophylla*, *Pinus ponderosa*, *Pseudotsuga menziesii*, *Quercus chrysolepis*, and *Quercus kelloggii* have seedling-only plots with significantly greater mean summer precipitation and significantly lower mean temperature of the warmest month than their corresponding tree-plus-seedling plots (two-sided $t$-test, $p < 0.05$). Furthermore, the average differences in these two climate variables across the five species are greater in plots that burned than in plots that did not (Fig. 4). For example, the difference in MSP between seedling-only and tree-plus-seedling plots is 2.2x greater in burned than unburned plots on average ($p \ll 0.05$). The climatic differences of these five species with regards to the other variables, MTCM and MWP, are less consistent. The three subalpine species, which do not exhibit lower MTWM or higher MSP in seedling-only plots, are similar to each other in that they all have higher MTWM and lower MTCM in seedling-only plots in both burned and unburned plots but no significant difference between the magnitude of climatic differences of burned and unburned plots (Supplementary Fig. 3).

**Contextualizing the direction of SORDs.** All four climate variables changed significantly across our study area between 1975 and 1995 based on 30-year climate averages (two-sided $t$-test, $p \ll 0.05$). Temperatures increased at a greater relative magnitude than precipitation decreased, and changes in precipitation

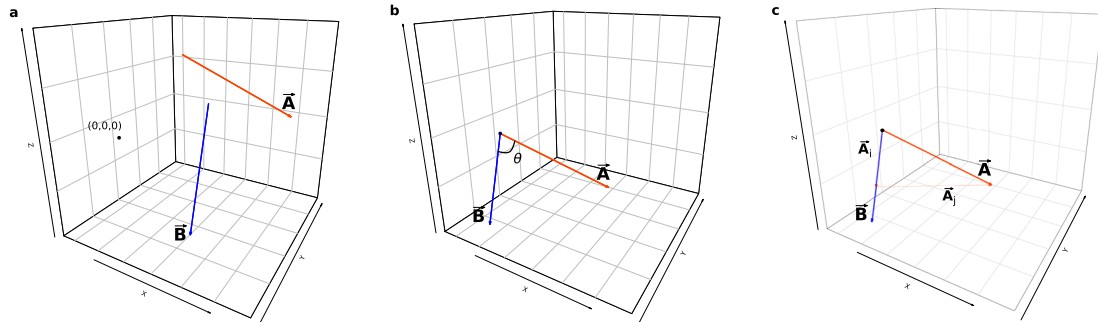

**Fig. 2 A synthesized example of the SORD vector-direction comparison methodology.** After scaling the environmental variables across the study area, we plotted the SORD vectors (exemplified here by $\vec{A}$ and $\vec{B}$) with the initial point of each vector at the centroid of the source population and the terminal point of each vector at the centroid of the seedling-only population (**a**). We then translocated the vectors to the origin and found the angle $\theta$ between them (**b**) and calculated the cosine of $\theta$ which is equivalent to $\frac{|\vec{A}_j|}{|\vec{A}|}$, i.e., the component of the normalized $\vec{A}$ that lies along normalized $\vec{B}$ (**c**).

**Table 1 Summary of the sample sizes and SORD vector agreement values that were used to determine the species included in further analysis.**

| | Number of plots | | | | | | SORD vector agreement | |
| | Burned | | Unburned | | | | | |
| | Trees & seedlings (BT) | Seedlings only (BS) | Trees & seedlings (UT) | Seedlings only (US) | Large tree & saplings (UG) | Saplings only (UJ) | Between life stages (US-UT vs. UJ-UG) (%) | Burned vs. unburned (BS-BT vs. US-UT) (%) |
|---|---|---|---|---|---|---|---|---|
| *Chrysolepis chrysophylla* | 27 | 25 | 452 | 539 | 134 | 869 | 82.6–92.7 | 95.0–98.2 |
| *Picea engelmannii* | 67 | 23 | 3122 | 772 | 3395 | 3277 | 69.9–85.1 | 97.2–83.8 |
| *Pinus albicaulis* | 23 | 7 | 843 | 430 | 721 | 1071 | 51.7–83.7 | 50.4–82.8 |
| *Pinus contorta* | 230 | 28 | 3701 | 340 | 3536 | 7107 | 68.6–89.2 | 75.8–89.2 |
| *Pinus ponderosa* | 115 | 30 | 2932 | 473 | 5273 | 2868 | 80.7–100 | 60.7–100 |
| *Pseudotsuga menziesii* | 350 | 29 | 10659 | 952 | 16088 | 5857 | 92.3–95.5 | 87.0–100 |
| *Quercus chrysolepis* | 149 | 43 | 1182 | 408 | 533 | 966 | 83.9–99.9 | 76.4–100 |
| *Quercus kelloggii* | 43 | 6 | 553 | 136 | 644 | 749 | 73.4–100 | 63.5–100 |

The agreement between different sets of SORD vectors where 100% indicates that the vectors share the same direction, and 0% indicates that the angle between vectors are greater than 90° include the standard error calculated from 5000 bootstraps.

**Table 2 Summary of the SORD metrics.**

| | Schoener's D | | Centroid distance | | Difference in centroid distance ($CD_B - CD_U$) | |
| | Burned | Unburned | Burned | Unburned | Mean | 95% CI |
|---|---|---|---|---|---|---|
| *Chrysolepis chrysophylla*[R] | 0.518* | 0.641* | 0.592 | 0.677* | −0.148 | (−0.795, 0.176) |
| *Picea engelmannii* | 0.369* | 0.751* | 0.508† | 0.375* | 0.132 | (−0.246, 0.406) |
| *Pinus albicaulis* | 0.320 | 0.768* | 0.294 | 0.234* | 0.0600 | (−0.416, 0.204) |
| *Pinus contorta*[s] | 0.534 | 0.692* | 0.308 | 0.314* | −0.00600 | (−0.356, 0.201) |
| *Pinus ponderosa* | 0.498 | 0.680* | 0.616* | 0.198* | 0.418 | (−0.0870, 0.724) |
| *Pseudotsuga menziesii* | 0.421* | 0.704* | 1.095* | 0.592* | 0.504* | (0.012, 0.964) |
| *Quercus chrysolepis*[R] | 0.688* | 0.852* | 0.714* | 0.282* | 0.432* | (0.109, 0.724) |
| *Quercus kelloggii*[R] | 0.574 | 0.825* | 0.641† | 0.194* | 0.447 | (−0.242, 0.916) |

The sample sizes used to produce these metrics are found in Table 1. Values appended with * or † indicate statistically significant evidence of SORD ($p < 0.05$ and $p < 0.1$, respectively), where the null hypothesis of no SORD corresponds to Schoener's $D = 1$ and centroid distance $= 0$ (*Chrysolepis chrysophylla*: $p_{Schoener's\ D,Burned} = 0.0199$, $p_{Schoener's\ D,Unburned} = 0.00398$, $p_{Centroid\ Distance,Burned} = 0.200$, $p_{Centroid\ Distance,Unburned} < 2.20e^{-16}$; *Picea engelmannii*: $p_{Schoener's\ D,Burned} = 0.00398$, $p_{Schoener's\ D,Unburned} = 0.00398$, $p_{Centroid\ Distance,Burned} = 0.0864$, $p_{Centroid\ Distance,Unburned} < 2.20e^{-16}$; *Pinus albicaulis*: $p_{Schoener's\ D,Burned} = 0.462$, $p_{Schoener's\ D,Unburned} = 0.00398$, $p_{Centroid\ Distance,Burned} = 0.372$, $p_{Centroid\ Distance,Unburned} = 6.66e^{-16}$; *Pinus contorta*: $p_{Schoener's\ D,Burned} = 0.251$, $p_{Schoener's\ D,Unburned} = 0.00398$, $p_{Centroid\ Distance,Burned} = 0.251$, $p_{Centroid\ Distance,Unburned} < 2.20e^{-16}$; *Pinus ponderosa*: $p_{Schoener's\ D,Burned} = 0.131$, $p_{Schoener's\ D,Unburned} = 0.00398$, $p_{Centroid\ Distance,Burned} = 1.13e^{-4}$, $p_{Centroid\ Distance,Unburned} = 9.10e^{-10}$; *Pseudotsuga menziesii*: $p_{Schoener's\ D,Burned} = 0.00398$, $p_{Schoener's\ D,Unburned} = 0.00398$, $p_{Centroid\ Distance,Burned} = 6.42e^{-4}$, $p_{Centroid\ Distance,Unburned} < 2.20e^{-16}$; *Quercus chrysolepis*: $p_{Schoener's\ D,Burned} = 0.00398$, $p_{Schoener's\ D,Unburned} = 0.00398$, $p_{Centroid\ Distance,Burned} = 9.08e^{-6}$, $p_{Centroid\ Distance,Unburned} = 1.65e^{-7}$; *Quercus kelloggii*: $p_{Schoener's\ D,Burned} = 0.725$, $p_{Schoener's\ D,Unburned} = 0.0159$, $p_{Centroid\ Distance,Burned} = 0.00847$, $p_{Centroid\ Distance,Unburned} = 0.00156$). Lower values of *Schoener's D* and greater Centroid Distance suggest greater SORD. $CD_B - CD_U$ is the result of subtracting the Centroid Distance in unburned samples by the Centroid Distance in burned samples, where the null hypothesis is that $CD_B - CD_U = 0$ and wildfire occurrence does not impact SORD. Statistical significance for this metric was calculated using a two-sided bootstrap test. Superscripts appended to the names of species indicate adaptations that facilitate post-fire regeneration, where [R] denotes resprouting capabilities and [s] denotes serotiny. See the discussion section for consideration on post-fire regeneration adaptations and observed SORD.
*Schoener's D* and centroid distance were tested for statistical significance using the one-sided niche equivalency test and two-sided Hotelling's $T^2$ test, respectively.

variables had a greater variance (Fig. 5). Comparing these recent climatic shifts in the study area to the SORDs of the eight tree species, most species that exhibit statistically significant differences show a trend in seedling-only plots towards historical climate values. For example, MSP decreased across the study area and five species have seedling-only plots with significantly higher values of MSP than tree-plus-seedling plots, consistent with a potential range shift towards regions of higher MSP. Only one species, *Picea engelmannii*, had seedling-only plots at lower MSP than tree-plus-seedling plots (Supplementary Fig. 2).

Variability in the relationship between the direction of SORDs and recent climate changes for each species loosely falls along the dichotomy of subalpine and non-subalpine species. The seedling-only plots of non-subalpine species exhibit a clear trend towards historical climate values for MSP and MTWM, with a greater trend (i.e., magnitude of SORD) in burned plots on average. On the contrary, the seedling-only plots of subalpine species have higher MTWM and MTCM, even though these temperature variables increased across the study area (Supplementary Fig. 3).

## Discussion

Our results contribute to the exploration of the hypothesis that forest fires can facilitate climate-induced range shifts[28,34], providing some of the first empirical evidence of contemporary climate-induced potential range shifts being facilitated by wildfire.

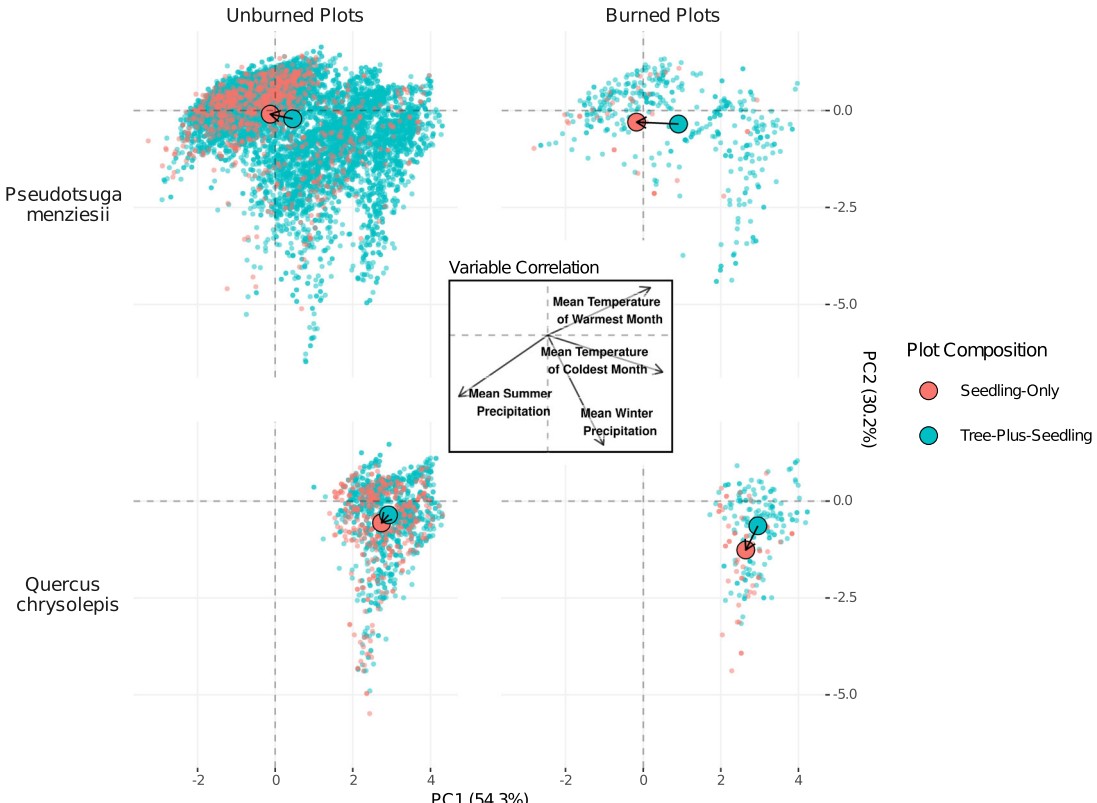

**Fig. 3 PCA of Climatic Niche Differences for *Pseudotsuga menziesii* and *Quercus chrysolepis*.** PCA plot of the climatic niches of seedling-only and tree-plus-seedling plots of the two species where SORD is greater in burned areas (Table 2). PC1 and PC2 explain 84.5% of the variation across all 4 climate variables. Centroids are shown as large points. Similar PCA plots for all species can be found in Supplementary Fig. 1. Source data are provided as a Source Data file.

Two of the eight species studied showed greater SORDs in burned areas than unburned areas, and these findings are consistent with current understandings of range shifts, climate change, and disturbance ecology. The greater magnitude of SORDs in burned areas could indicate that competition can constrain range shift rates as tree species respond to climate change.

Three sources of uncertainty limit the strength of this interpretation (i) potential non-maturing populations where seedlings occur but have not, and may not, reach maturity (ii) fire succession and fire-regime adaptations and (iii) the relationship between climatic distance and spatial distance (Eq. (1)).

Our approach was based on the postulate that plots with seedlings only are the leading edge of a range shift that becomes complete when the seedlings eventually mature into trees. This is, of course, not necessarily the case. We excluded 14 species where there was an indication that there were non-maturing populations (Table 1 and Supplementary Table 1), by comparing the direction of observed SORDs to the climatic niche displacement between saplings (2.5 cm < dbh < 12.7 cm) and large trees (dbh > 12.7 cm). This screen should have excluded species from further analysis that had an abundance of non-maturing populations, although, it is still possible that some saplings of the remaining 12 species were members of non-maturing populations and would never reach reproductive maturity.

Fire adaptation and succession dynamics could confound the results where SORDs in burned areas were entirely controlled by post-fire colonization adaptations. We minimized the likelihood of this by excluding 4 species which exhibited inconsistent SORD directions between burned and unburned plots, which we interpreted as evidence for SORDs driven more by the response of a species to wildfire than to climate change. The SORDs of the remaining 8 species were in the same direction in burned and unburned plots, a pattern inconsistent with responses to different drivers in the burned and unburned plots. Figure 2 provides a graphical explanation of the SORD vector vetting, and is useful for understanding the relationship between SORD and fire adaptation driven displacement. This method has the additional advantage of screening out species where wildfire might drive SORD for reasons less-related to post-fire colonization adaptations, such as fire-induced mortality of mature trees. See Supplementary Table 1 for the SORD metrics of species that were filtered out by this analysis.

The nature of the FIA data precludes a deep exploration of the relative contributions of fire's direct, abiotic impacts and its climate-change mediated, biotic impacts. The FIA fire data provide invaluable geographic accuracy, but the limited description of fire characteristics prevents an in-depth investigation of the mechanistic interactions between wildfire impacts and potential range shift rates. FIA data collectors recorded only fires that caused damage/mortality to more than 25% of standing trees, then classified these fires as either crown or ground fires (the "ground fire" categorization refers to surface fires). Ground fires (63% of the fires cited across the FIA plots of this study) are typically of lower intensity than crown fires and can lead to increased soil nutrients and microbial activity, while moderately increasing erosion. Higher intensity fires (of which many are crown fires) can, but not necessarily, cause soil sterilization, nutrient depletion, and greater erosion[48]. The ecological impacts of these various types of fires are further complicated by the response of each species to the varied impacts of wildfire.

Some of the species in this study have traits that make them particularly well-adapted to regenerate/colonize after fire.

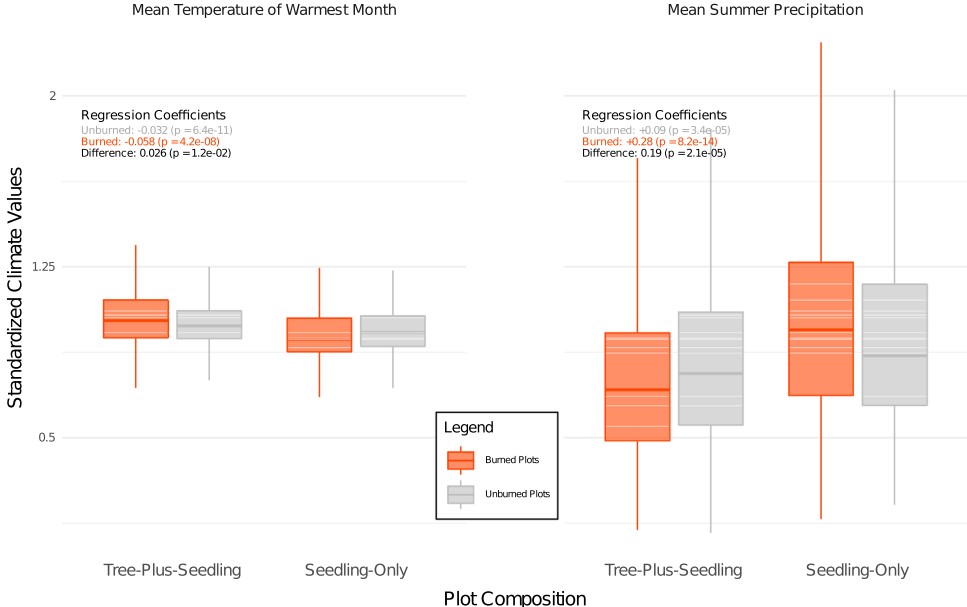

**Fig. 4 The climatic components of SORD most impacted by wildfire occurrence, separated by wildfire history, and aggregated across species.** The plot includes results for all species that share the direction of a statistically significant difference (two-sided *t*-test, *p* < 0.05) between seedling-only (*n* = 1806) and tree-plus-seedling (*n* = 5654) populations in unburned plots. The species included are: *Chrysolepis chrysophylla*, *Pinus ponderosa*, *Pseudotsuga menziesii*, *Quercus chrysolepis*, and *Quercus kelloggii*. Boxplots include the median line, a box denoting the interquartile range, and whiskers showing values ± 1.5x the interquartile range. This figure demonstrates that while these five species show a trend in unburned seedling-only plots towards lower mean temperature of the warmest month and higher mean summer precipitation, the difference in burned plots is greater (*p* < 0.01). Climate variables were standardized by dividing the values by their root-mean-squares. Multiple linear regression was used to quantify the difference in SORDs between burned and unburned samples. Supplementary Fig. 3 shows this analysis across the full suite of climate variables, including those where the difference between the SORDs of burned and unburned samples were not statistically significant. Source data are provided as a Source Data file.

Resprouting from vegetative buds is a common post-fire regeneration strategy implemented by *Quercus chrysolepis*, *Quercus kelloggii*, and *Chrysolepis chrysophylla*[49–51]. FIA data do not distinguish between resprouts and seedlings, and some resprouts may have been misclassified as seedlings in our analysis. However, this would be unlikely to inflate the calculated SORD of these species because resprouts would never satisfy the seedling-only classification: the presence of seedlings and absence of living or dead mature trees. Resprouting would inflate the SORD of a species only in the unlikely case that some plots contained only resprouts of that species, and the mature trees were burned beyond species-level identification, and if the burned plots were preferentially in the same direction in climate space as the SORD in unburned plots. Similarly, the serotinous cones (which open and release seeds after being heated by fire) of *Pinus contorta* subsp. *latifolia* could increase this taxon's colonization rate into burned plots[52], but this would inflate the observed SORD only if the burned plots were preferentially in the same direction in climate space as the SORD in unburned plots. In addition, *P. contorta* subsp. *latifolia* is one of three *Pinus contorta* subspecies within our study area—distributed across the Rocky Mountains and northern Cascades[52]. The non-serotinous subspecies *Pinus contorta* subsp. *murrayana* and *Pinus contorta* subsp. *contorta* also cover a significant range within the study, with *murrayana* covering the Sierra Nevada and much of the Cascade Mountains[53] and *contorta* distributed along the coast from Northern California through Washington[54]. While it is likely that populations with fire-adapted traits across our study area increased the ability of some species to colonize after fires, it is unlikely that these traits shifted the SORD of burned plots in the same direction as the potential range shift in unburned plots. The effect of fire-adaptations on SORD is further complicated by their dependence on the characteristics of the fire. For example,

Rodman et al. found that *Pinus ponderosa* was better than *Pseudotsuga menziesii* at recolonizing after smaller, low severity fires but posited that *Pseudotsuga menziesii* would be better at colonizing after larger fires because of its longer dispersal distance[55].

The FIA recorded only fires that occurred within 5 years prior to the survey. Fires that burned 6 years prior could have affected the vegetation regeneration/recruitment (and therefore SORD) but would have been included in the unburned rather than burned tree-plus-seedling and seedling-only groups. This would have homogenized the observed SORDs between burned and unburned groups and decreased the probability of detecting an impact of fire on SORD. This implies that, with a more comprehensive ability to identify burned plots, the effects of fire on SORD would probably have been even larger.

Fire also has indirect, biotic impacts. Pervasive across the permutations of wildfire's ecological impacts is the increased availability of niche space. In general, where resources are more abundant (like light and nutrients) and competitors fewer, populations grow. Following a fire, population establishment and growth is accomplished by recolonization from adjacent populations, unless species have particular traits that aid in post-fire regeneration (like the aforementioned serotinous cones or resprouting). A gap in the canopy or large high-severity burn patch potentially promotes establishment through reduced competition. The species that colonize and establish in these sites are determined by myriad factors some stochastic, others intimately contingent upon the relationship between plot characteristics and species traits. We expect that the potential leading-edge populations (seedling-only plots) of species observed in this study had a competitive advantage colonizing and establishing in the more burned and depauperate locales where the climate had shifted to become more similar to the colonizers' climatic niche. In other

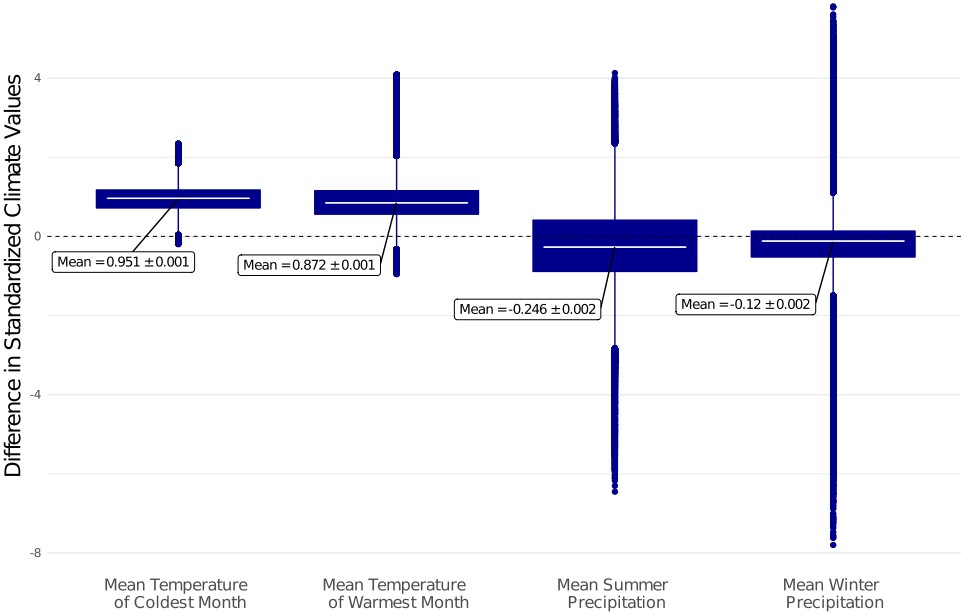

**Fig. 5 Recent climate change over the study area.** Difference between average 1981–2010 climate and average 1961–1990 climate across the study area ($n_{grid\text{-}cells} = 886{,}911$), showing 95% confidence interval and range (standardized by Z score). Boxplots include the median line, a box denoting the interquartile range, whiskers denoting values ±1.5x the interquartile range, and points denoting all outliers. Changes in all climate variables were statistically significant (two-sided $t$-test, with $p < 2.2e^{-16}$ for each climate variable). The temperature variables increased at a greater magnitude than the precipitation variables decreased. Source data are provided as a Source Data file.

words, in areas where the climate had changed, species adapted to the new climate had a greater chance of establishing in recently burned areas. While this study provides clear support for a role of fire in facilitating range shifts consistent with climate change, the evidence is still incomplete. Further work, on these and other ecosystems, will be necessary to provide a comprehensive understanding of the interactions between post-fire succession and climate-induced range shifts and their effect on post-fire community assembly (see Coop et al.[36] and Davis et al.[56] for more information on the state of this endeavor[36,56]).

The amount of climate change in the Coupled Model Intercomparison Project (CMIP) data across our study area is consistent with the observations and projections of other studies, particularly with regards to decreased summer precipitation and mean temperature increase[57,58]. Precipitation and temperature variables are among the most important for determining plant distributions, and these variables can affect plant traits/distributions in concert and compound each other's effects. Five of eight species shared a similar relationship to both MSP and MTWM—*Chrysolepis chrysophylla*, *Pinus ponderosa*, *Pseudotsuga menziesii*, *Quercus chrysolepis*, and *Quercus kelloggii* all had seedling-only plots with higher MSP and lower MTWM than their respective tree-plus-seedling plots. Lower MSP and higher MTWM are both associated with drought-stress in trees, which can lead to mortality[59–61]. Based on the available information, the trend of the seedling-only plots of these five species towards lower MTWM and higher MSP could reflect an increase in suitability for the advancing seedlings or a decrease in suitability for the previous dominants, or some combination of the two. The available evidence does not address the relative roles of the two mechanisms.

The three species that did not have seedling-only plots trending towards higher MSP or lesser MTWM were *Pinus contorta*, *Picea engelmannii*, and *Pinus albicaulis*, the only three species in this study that populate subalpine regions. This could reflect the constraint that limited potential establishment sites at the upper-elevation margins of their distributions leads intrinsically to a greater abundance of sites with suitable substrate at the lower-elevation margins and thus a greater abundance of seedling-only plots. This pattern, which is likely independent of climate change, provides a cautionary note about interpreting every instance of differences in seedling-only and tree-plus-seedling distributions as indicating a range shift in progress.

The observed SORDs, when roughly converted to spatial distance, are also consistent with general expectations. Using the normal lapse rate of 6.5 °C/km, we estimate that the observed MTWM component of SORDs for non-subalpine trees corresponds to an altitudinal displacement of +99 m in unburned plots and +190 m in burned. The difference in MTWM of sub-alpine species corresponds to −100 m and −110 m altitudinal displacement for unburned and burned plots, respectively. These values are within the range of altitudinal range shifts observed in western Europe, where trees shifted up to 214 m in mean elevation over the last century[62].

Some uncertainty comes from using climatic distance as a proxy for spatial distance (Eq. (1)). The relationship is certainly not fixed, particularly across topographically diverse regions or fragmented habitats. A 400 m horizontal distance will likely be much more climatically distant in a mountainous than a flatter region. Climatic distance is important in itself. Further, climatic and spatial distance are strongly correlated[12]. One possible confounding effect would occur if burned plots were on steeper slopes. In this case, larger climatic distances in burned plots might not correspond to larger spatial distances. We found that slopes are significantly steeper in burned (mean = 23.6°) than in unburned plots (mean = 20.2°) according to a two-sided $t$-test ($p \ll 0.05$) (Supplementary Fig. 4). Steeper slopes may have decreased the ratio of geographic distance to SORD in burned plots, potentially explaining a fraction of the estimated geographic displacement that corresponds to SORD. At a slope of 20.2°, the 99 m altitude displacement of the unburned plots of non-subalpine species corresponds to a lateral geographic

displacement of 269 m. On the same slope, the 190 m altitude displacement of burned plots would correspond to 516 m, but since the greater slope of 23.6° makes this lateral geographic displacement 435 m we might expect that the ratio of lateral geographic displacements for burned and unburned plots could be up to $\frac{516m-435m}{516m-269m}*100\% = 32\%$ smaller than the ratio of climatic niche displacements. This estimate, based on the MTWM component of SORD alone, illustrates the potential complexity in translating from one aspect of climate space to geographic distance and does not attempt to capture the additional information in the 4-dimensional climate space.

This study provides empirical evidence that wildfire can increase the potential range shift rate of tree species that are moving in response to recent climate change. Furthermore, this pattern supports the argument that range shift rates are likely affected by reductions in population size and density of competitors due to wildfire. The findings of this study bolster previous work suggesting that competition is yet another barrier (in addition to dispersal limitations and geographic barriers) that affects the ability of plant species to track their optimal climatic conditions as they move across landscapes.

## Methods

**Occurrence data.** Plot-level tree and seedling species lists were sourced from the USDA Forest Inventory and Analysis (FIA) program via the FIA DataMart tool (v 8.0, https://apps.fs.usda.gov/fia/datamart) in April of 2021 and analyzed using R (v 4.1.0)[63] and RStudio (v 1.1.463)[64]. The FIA data also include tree age, tree diameter at breast height (dbh), and the presence of fire disturbance at the plot (fires must have occurred, at most, 5 years prior to plot establishment and caused damage and/ or mortality to at least 25% of trees over an area greater than 0.4 ha to have been recorded). FIA datasets are the most comprehensive continent-scale source for plot-level forest data in the U.S. In FIA Phase 2 data collection (sourced in this study), a plot is divided into four 168 m² subplots and four 13.5 m² microplots. Within subplots, all tree species (living and dead) with a dbh greater than 12.7 cm are identified and tallied, and all seedlings are identified and tallied within microplots. FIA data do not distinguish between seedlings and resprouts and tally multiple suckers from the same root system (must be from a dead tree) as one seedling[65].

We sourced all plots (74,069 total) within the Northwestern Forested Mountains and Marine West Coast Forest ecoregions of the continental U.S.[66] (Fig. 1). All plots used in this study were surveyed between 1999 and 2019. For each species we separated the plots into unburned (control) and burned (treatment) groups. Within these two groups, novel seedling establishment was used as a proxy for range shift as outlined in Zhu et al.[10]. Seedlings in plots with no adult tree (hereafter referred to as "tree") of the species were considered the potential leading-edge of the migrating population. Subsequently, the FIA plots of each species were separated into 4 independent groups: (1) Plots that did not burn in the last 5 years and had only seedlings present at the time of sampling (unburned, seedling-only); (2) Plots that did not burn in the last 5 years and had seedlings and trees present at the time of sampling (unburned, tree-plus-seedling); (3) Plots that burned in the last 5 years and had only seedlings present (burned, seedling-only); (4) Plots that burned in the last 5 years and had seedlings and trees present (burned, tree-plus-seedling)). Species were removed from analysis if the number of plots was less than 5 in any of these four categories. This criterion removed 84 species.

Because of the spatial uniformity of FIA sampling, spatial sampling bias is unlikely to be a major source of error in our analysis. However, seedlings were sampled in only a subset of each FIA plot, and it is possible that seedlings present in the full sampling plots were absent in the seedling microplots, potentially reducing the number of burned and unburned seedling-only plots.

**Climate data.** Rasterized climate data (averaged from 1981 to 2010) were sourced at 30 arc-second resolution from the 2015 AdaptWest Project (https://adaptwest.databasin.org), constructed using ClimateNA v5.10 software[67]. These climate data cover a period that ended nine years prior to the most recent FIA sampling. We initially considered 8 possible climate variables, which were used for similar studies and recommended for climatic niche analysis because of their relevance to plant physiology: Mean Temperature of the Coldest Month, Mean Temperature of the Warmest Month, Mean Annual Precipitation, ratio of actual to potential evapotranspiration, potential evapotranspiration, precipitation seasonality, mean annual temperature, and growing degree days. We replaced precipitation seasonality with its seasonal components, mean summer precipitation and mean winter precipitation, to make the results more interpretable and to isolate shifts in precipitation regimes across the study area, which varies (summer precipitation, particularly) across mountain, coastal, and mediterranean climates. We calculated the Variance Inflation Factor (VIF) for the set of 9 variables using

the R package *usdm*[68] and incrementally excluded collinear variables until VIF < 10, as recommended, and used the 4 climate variables with the least collinearity (see Supplementary Table 2 for more information). The resulting climate variables were: Mean Temperature of the Warmest Month (MTWM), Mean Temperature of the Coldest Month (MTCM), Mean Summer Precipitation (MSP), and Mean Winter Precipitation (MWP).

**Species selection.** The climatic difference between seedling-only plots and tree-plus-seedling plots may not represent the actual range shift direction of a species. For example, seedling establishment may represent non-maturing populations that occur in areas that will not support a reproducing population in the future. To ensure that we included only species for which we could confidently estimate the direction of potential range shift, we compared the Seedling Only Range Displacement (SORD) vector of unburned tree-plus-seedling plots and seedling-only plots with the same vector adjusted to reflect the displacement of unburned sapling-only plots (with dbh between 2.5 cm and 12.7 cm) from unburned large tree plots, with dbh greater than 12.7 cm[10]. We determined the directional consistency of the niche difference between the different life stages by calculating the component of the seedling niche difference vector, $\vec{ND}_{seed}$ (i.e. SORD), that pointed in the same direction as the sapling niche difference vector, $\vec{ND}_{sapling}$, i.e., the dot product of the normalized niche difference vectors in climate space, $\frac{\vec{ND}_{seed}\vec{ND}_{sapling}}{|\vec{ND}_{seed}||\vec{ND}_{sapling}|}$. Figure 2 provides a graphical explanation of this process. We used the cutoff of 50% vector agreement to exclude species where the niche difference direction was not consistent across life stages.

The direction of the SORD vectors may differ significantly between the unburned and burned plots, possibly because of species-specific adaptations that affect colonization/regeneration after fire. To limit the possibility of range displacements driven by fundamentally different mechanisms in burned and unburned plots, we also excluded species for which the cosine of the angle between SORD vectors in burned and unburned plots was less than 0.5 (Fig. 2).

**Calculating SORD.** We used two metrics to calculate SORD. Schoener's D is used to evaluate whether the difference in climatic niches between tree-plus-seedling and seedling-only plots are significantly different. However, this metric lacks directionality. Euclidean centroid distance provides an alternative metric for describing SORDs, but, with both a magnitude and direction, it also allows us to evaluate the difference in SORDs between burned and unburned samples.

Schoener's D was calculated using a modified R script[69] that calculates the two principal components which describe the most variation in climate space between tree-plus-seedling and seedling-only plots, and then measures the difference between the kernel-smoothed component scores for the two sets of plots. The statistical significance of Schoener's D is determined with the niche equivalency test of Broennimann et al.[69].

Euclidean centroid distance between the climatic niche centroids of seedling-only and tree-plus-seedling plots more closely approximates a potential range shift rate (Eq. (1)).

$$r_i = \frac{DG_i}{t} \approx \frac{DC_i}{A_{Ti} - A_{Si}} \tag{1}$$

Where $r_i$ is the potential range shift rate and i is either burned or unburned, DG is the geographic range shift distance per unit time (t), DC is the climatic niche distance per unit time, where time is the mean age difference between mature trees ($A_T$) and seedlings ($A_S$). The Euclidean distance between the centroids for unburned tree-plus-seedling and unburned seedling-only plots and between burned tree-plus-seedling and burned seedling-only plots (derived from the magnitude of the niche difference vectors described above) give DC$_{Unburned}$ and DC$_{Unburned}$, respectively. We do not have information on $A_T$ and $A_S$, but we have no reason to expect them to be different between burned and unburned plots. Therefore, we use the difference between CD$_B$ (centroid distance, burned) and CD$_U$ (centroid distance, unburned) as our primary measure of whether the magnitude of SORDs are different between burned and unburned plots. We bootstrap the calculation of CD$_B$–CD$_U$ using 20,000 replicates to produce 90% and 95% confidence intervals.

**Continued SORD analyses.** We continued our analysis of SORDs by breaking the multi-dimensional values back into their component variables (e.g., MSP, MTWM, etc.). Because we were interested in comparing the SORDs between burned and unburned plots for each climate variable (rather than for each species, as performed earlier) we aggregated plots across species to maximize the sample size of plots in this analysis. We aggregated the plots of only species for climate variable $C_i$ where we could demonstrate that the unburned SORD for $C_i$ was significant (two-sided t-test, $p < 0.05$) and pointed in the same direction (e.g., towards higher MSP or lower MTCM). For example, the plots of *Pinus contorta* and *Pinus albicaulis* would be aggregated for the analysis of MSP if the unburned plots of these species independently demonstrated a shared seedling-only range displacement towards higher MSP. When aggregating the plots of species we used stratified sampling to ensure that the sample size was equivalent between species, and averaged the results across

100 sampling events. We used multiple linear regression to evaluate the difference in climatic distances between burned and unburned plots for each climate variable.

**Evaluation of methodological robustness**. To test the robustness of our results to different sets of climate parameters, we compared the results presented in this manuscript with a suite of results obtained from the use of different sets of climate variables. We reanalyzed our data with additional, though not exhaustive, combinations of climate variables with VIF < 10 (Supplementary Table 2). The 3 alternative sets of results did not differ appreciably from our primary results— Schoener's D and Euclidean centroid distance consistently indicated SORDs were greater in plots that burned for a specific set of species (Supplementary Tables 3.1–3.3).

To ensure that the threshold minimum sample size did not greatly impact the results, we repeated our analysis of range shift rates for minimum sample size of presence in 5, 10, and 25 FIA plots. The threshold minimum sample size affects the number of species analyzed but not the conclusion that wildfire occurrence is correlated with greater SORD for some species (Supplementary Table 4). Our primary analysis uses a minimum sample size of 5 because two species with sample sizes below 10 were not filtered out during the SORD vector direction vetting, indicating that the minimum sample size of 5 was sufficient to estimate a SORD vector that was consistent with populations of larger sample sizes. We also verified that other decisions regarding species exclusion, such as species vetting due to SORD vector agreement between life stages and between burned and unburned plots, did not produce results that contradicted our primary conclusions (Supplementary Table 1).

**Reporting summary**. Further information on research design is available in the Nature Research Reporting Summary linked to this article.

## Data availability
The climate and plant occurrence data used to support the findings of this study are openly available from the AdaptWest Project and Forest Inventory Analysis, respectively. Source data are provided with this paper.

## Code availability
R scripts used and produced in this study can be found at the GitHub repository: https://github.com/avephill/wildfire-plant_RS[70].

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

## Acknowledgements

A grant from the Gordon and Betty Moore Foundation (ID: GBMF8449) to C.B.F. facilitated completion of this work. We would like to thank those that provided prompt and invaluable feedback on the manuscript, including Robert S. Harbert at Stonehill College, Kai Zhu at UC Santa Cruz, and Lucas Pavan at Stanford University. A.P.H. would also like to thank Raymond and Neta Thornell for their enduring and consequential support.

## Author contributions

A.P.H. and C.B.F. designed the thesis and research jointly, and A.P.H. conducted the data analysis and wrote the first draft of the manuscript. Both authors worked on subsequent revisions to both methodology and manuscript.

## Competing interests

The authors declare no competing interests.
