## [Peer Review File · Nature Communications]

Reviewer comments, initial round review:

Reviewer #1 (Remarks to the Author):

I have read the revised version of "How Competition and Wildfire Affect Tree Range Shifts in the American West". I was a reviewer of the original submission to Nature Climate Change. I am very impressed with the thorough revision accomplished by the authors based on my and two other reviewers' comments. In particular, I think the authors have done a good job of overhauling the Discussion, as well as reworking their wording so as not to overstate competition as a certain/sole mechanism of the increased range shifts. I also believe they have assiduously addressed all of the caveats and nuances to their methodological approach, while not letting those caveats take away from the impact of the findings (which the first version suffered from). I also appreciate the work they did to link some estimate of spatial movement (using the lapse rate) to the observed shift in climate space.

A few very small comments:

Title: I suggest changing "the American West" to the western United States, as the Americas include North and South America.

Lines 177-178: I suggest changing to "but none are as common and severe in western North America as wildfire" to "but none are as common and severe in the western United States as wildfire". Otherwise, you are claiming this statement for all of North America, which is probably not the case.

Line 212: Change "welcomes" to "facilitates" or "promotes"

Reviewer #2 (Remarks to the Author):

See attached review.

Reviewer #3 (Remarks to the Author):

As far as I know, this is the first empirical test of the hypothesis proposed (which was suggested earlier by Pausa & Keeley 2019). Despite some methodological shortcomings (mostly recognised by the authors), the results are interesting and the manuscript could make a significant contribution to the topic. However, in the current version, some of the statistics and results are unclear, and the text needs to be much improved. The Discussion section is too long (some parts can be condensed, others moved to Methods); and tables and figures are poor (in the presentation and in the details of the caption). Below I provide some specific comments.

Title: Remove 'competition and' as you are not measuring competition; competition is inferred as possible mechanism.

Line 26, etc. - but see papers by W. Bond (e.g., 2005 in New Phytol, 2020 in Trends Plant Sci, etc.), where he and col. show that plant biogeography is not only determined by climate (your paper is another example).

Lines 60-61, 72-73, 164, ... You should mention that this hypothesis was proposed earlier (Pausas & Keeley Front. Ecol & Environ.), and your objective is to test it.

Line 78: assuming that the presence of seedlings is a "leading edge" is a weak point.

Table 1; how you (or FIA) differentiate between oak seedlings and oak resprouts?

Lines 117-122. Where are these results? I do not see DC and DCb/DCn values in the tables and

figures.

There are terminological problems that makes the text confusing; e.g., the "treatments" are termed in different way in the text (Burned/Unburned), in the tables (Inside/Outside fire), and in the acronyms (B, N). Are the "Novel seedlings" in Table 1 the "Seedlings only" in the text? What do 'NJ' and 'NG' mean in Table 1? Please take care and be consistent with the terminology used through the text, and include the meaning of the acronyms in tables (so they can be understood by their own).

Table 2. * indicates significance of what? Different from 0? comparing Inside vs Outside?

Fig. 3a. Please add the statistical significance between Burned and Unburned plots for each variable. I suppose that Fig. 3b only makes sense in the case that the difference between B and U (in 3a) is significant.

Lines 132-133: The values 100% and 208% are a bit surprising, perhaps because they are in %. It should be better explained. If these are important values, then a table (in the Suppl. Mat.) for each species and variable (with more details on how they were computed) would help.

Line 136: 'range of RS' perhaps 'variability of RS'?

Line 142: 'faster' is this significant?

Line 164: add Pausa & Keeley (above).

Line 171: well, there are many examples of natural selection after 1 or 2 generations; the examples by Grant & Grant (beak sizes of finches) are a classical example, but there are many others.

Line 182-184: the concept of "climax" is a bit outdated now, especially in a paper that have a dynamic view of nature as this one. See papers by Bond (e.g., Trends in Pl Sci) on state-shifts.

Line 188, 235-253; these two species have been studied in detail by Rodman et al. (2020 J Ecol.) in relation wildfire responses; how your results fit with those in that paper?

Lines 193-201. Could be condensed.

Line 200: by ground fires do you mean surface fires? Understory fires?

Lines 203-4: High intensity (crown) fires are not always as negative as it seems here (sterilization, erosion, ...); it is likely a natural fire regime in the American boreal forest (Archibald et al. *Envir. Res. Lett.*). In fact some of your species may benefit from high intensity fires (is your *Pinus cortata serotinous*?).

Line 210: again, is any of your species serotinous? He et al. *New Phytol* may help. And do they resprout? Probably *Populus tremuloides* and *Quercus chrysolepis* are good resprouters. How this affects your results?

Line 213: "locales" → "microsite"?

Lines 235-245: see Rodman et al. (2020 J Ecol.)

261-265, methods

263-267: to add the info in the Suppl Mat.

Line 264: "eight trees" → "eight species"? or "eight tree species"?

The Discussion needs to be shortened and written in a more condensed way.

I would say that Fig. 5 is not mentioned in the main text, only in the methods, if so, it would be better to move this figure to the Suppl. Mat.

Fig. A1, the symbols differ between the plot and the legend.

REVIEWER COMMENTS

Reviewer 1

I have read the revised version of “How Competition and Wildfire Affect Tree Range Shifts in the American West”. I was a reviewer of the original submission to Nature Climate Change. I am very impressed with the thorough revision accomplished by the authors based on my and two other reviewers’ comments. In particular, I think the authors have done a good job of overhauling the Discussion, as well as reworking their wording so as not to overstate competition as a certain/sole mechanism of the increased range shifts. I also believe they have assiduously addressed all of the caveats and nuances to their methodological approach, while not letting those caveats take away from the impact of the findings (which the first version suffered from). I also appreciate the work they did to link some estimate of spatial movement (using the lapse rate) to the observed shift in climate space.

A few very small comments:

Title: I suggest changing “the American West” to the western United States, as the Americas include North and South America.

We maintain that “American West” sounds better enough to warrant a modest reduction in accuracy, as compared to “western United States.”

Lines 177-178: I suggest changing to “but none are as common and severe in western North America as wildfire” to “but none are as common and severe in the western United States as wildfire”. Otherwise, you are claiming this statement for all of North America, which is probably not the case.

This sentence was removed in the process of making the introduction clearer and more succinct.

Line 212: Change “welcomes” to “facilitates” or “promotes”

Changed to ‘promotes’.

Reviewer 2

This manuscript is a resubmission that included responses to previous reviews, but this was my first view of the paper. I read the revised manuscript and formed my opinions before reading the original reviews and authors’ responses. I find the topic highly relevant and I see great potential for utilizing the FIA dataset to address questions about fire-facilitated range shifts. As executed and written, however, my opinion is that the current manuscript falls short of this potential. Overall, my comments strongly mirror several key critiques and concerns raised in the original reviews, suggesting that the authors were unable to address some fundamental limitations or shortcomings in this revised manuscript. Below, I highlight some of my overarching concerns, which ultimately undermine the inferences, conclusions, and broad relevance of the work.

1. The three processes that are the focus of this paper – competition, wildfire, and range shifts – are either not measured at all, or measured in a very limited, indirect way. While these limitations all seem surmountable individually, for example through changing the language and

framing of the work, the current manuscript leans very heavily on these concepts through the framing, study design, and interpretation in the discussion.

Competition: The mechanism of reduced competition as an explanation for the patterns highlighted is largely assumed, as a uniform consequence of fire across all plots. Fire certainly alters the competitive environment for post-fire regeneration – and the revised discussion includes some general text on how competition for light, nutrients, and water can be reduced for conifer seedlings in the post-fire environment. However, fire can also increase competition (e.g. for space, water, and light, if grass or shrub species establish before conifers (e.g., Tepley et al. 2017), and fire can facilitate conifer regeneration through other mechanisms, including providing mineral soil/substrate for seedling establishment, and altering micro climate conditions (by removing canopy buffering). Maybe all of this could be grouped into the category of “competition,” (but probably not changing substrate)...but the logic and discussion comes off as highly forced into the framework of competition, with limited discussion and/or evidence of the assumed mechanisms.

My comments here largely echo original review #2, which noted “How is the competition aspect actually tested?” The response to that comment was that “...while this study does not prove competition affects...range shifts, the evidence is consistent with this hypothesis.” I would suggest that the bar of “proof” is much higher than any single study could hope to provide – so that’s not the issue. To the extent that these data are consistent with the hypothesis that fire facilitates range shifts via reduced competition, the data are also consistent with a number of other alternative hypothesized mechanisms. My opinion here echoes that of original review #3, which noted “...certainly there are many other abiotic and biotic changes between burned and unburned sites, and these results cannot be attributed causally to competition.” By focusing so strongly on competition (from the title on), there is an implicit assumption that the data confidently point to competition as the mechanism for the patterns documented. This is simply not well supported by the data analysis.

A limitation of this work is that we are unable to provide insight into the mechanisms by which fire led to the observed differences in range shift rates. We’ve overhauled much of the discussion and some of the intro to try and make explicit that we (1) are unable to parse the relative contribution of different drivers of the observed difference in range shift rate and (2) what this paper aims to do, instead, is show that wildfire affects range shift rate of different species, with reduction in competition as one, of many, drivers. Through our experimental design we do not prove that competition impacts range shift rate, rather, we attempt to show that our results cannot let us reject the hypothesis that competition affects range shift rates. We make the case that if we did not observe a relationship between fire and range shift rates, it would be very unlikely that any other reduction in competition would affect range shift rates. This is because we exclude species where the difference in range shift rates is *entirely* attributable to fire, so at least one of the drivers is climate change and increased niche space.

(Lines 67-70) “While the presence of an effect of fire in accelerating range shifts does not definitively prove a role for competition (the effect of fire on community assembly certainly

goes beyond the reduction competition), the absence of an effect of fire would provide strong grounds for rejecting the hypothesis.”

(Lines 202-206) “Fire adaptation and succession dynamics could confound the results where rates of range shifts into burned areas were entirely controlled by post-fire colonization adaptations, with no evidence of climate change as a driver. We minimized this effect by excluding species where the direction of range shift in burned areas was inconsistent with the direction of range shift in unburned areas.”

(Lines 210-211) “The nature of the FIA fire data precludes a deep exploration of the relative contributions of direct, abiotic impacts and the climate-change mediated, biotic impacts of fire.”

Fire: While the FIA dataset is a treasure trove of information on forest dynamics and disturbance, the treatment of “fire” is very simplistically, from the experimental design through the analysis. The study design hangs on the delineation of “burned” and “unburned” plots, but this oversimplifies the presence/absence of fire, and fire effects. Fire behavior and fire effects across this broad study region are extremely diverse. While text in the discussion (199-203) argues that most plots burned in low-intensity fires, which are assumed to have increased soil nutrients, etc., the text also notes that “63% of the fires” were ground fire. This leaves 37% (over 1/3rd) of the plots (presumably) as experiencing some level of passive or active crown fire activity. This variability in fire behavior (surface vs. crown) would likely translate into variable fire effects on both biological and physical aspects of the post-fire environment; one needs some inference into these effects to propose causal mechanisms for the patterns highlighted in the study. The methods (L 309) also note that fires in burned plots “caused at least 25% tree mortality over at least 0.4 ha” – this level of tree mortality typically gets classified as a moderate- or mixed-severity fires (and thus this contradicts the argument in the discussion that most fires burned at low intensity). The larger point here is that fire behavior and fire effects vary widely, but this variability is not well accounted for in this study, either explicitly or conceptually; this is problematic mainly because the interpretation of the data assumes a pretty narrow range of fire behavior and uniform fire effects.

Equally important, plots classified as “unburned” are really plots that have a time-since-fire of > 5 yr. While most conifer regeneration occurs within five years after a high-severity fire (e.g. Tepley et al. 2017), the ecological impacts of fire last much longer than five years (e.g., for decades to centuries). Thus, treating all plots with no evidence of fire for the previous five years as “unburned” is a major confounding factor in inferring the role of fire in facilitating range shifts. Given the total area burned over the 20th and early 21st century across this region, fire affected many of the “unburned” plots, it’s just a question of when. There are publically available datasets that could be used to account for fire activity across this entire region (and aspects of fire severity) back to 1984, which could help refine the experimental design (e.g., MTBS.gov; and see Parks et al. 2018).

You've helped illustrate the mismatch between our discussion of the ecological impacts of fire and the fire data and analysis we provided. We considered sourcing more descriptive fire data but concluded that the FIA fire data were uniquely suited for our analysis because of the paired sampling and spatial precision. We acknowledge that this choice limits our ability to provide any mechanistic understanding of the impacts of fire on range shifts, and have accordingly edited the discussion section (particularly Lines 210-221) to acknowledge the limitations of the data and the potential diversity of fire impacts that *may* have occurred.

The 5-year observation limit on fire occurrence is certainly another limitation of these data. However, we do not think this greatly confounds our results because all plots misclassified as 'unburned' that did in fact experience a fire within ecologically appreciable time would result in more homogeneity between the 'burned' and 'unburned' groups and therefore an underestimation of the impacts of fire on our calculated range shift rate.

(Lines 237-243) "The FIA recorded only fires that occurred within 5 years prior to the survey. Fires that burned 6 years prior could have affected the vegetation regeneration/recruitment (and therefore range shift rate) but would have been included in the NS and NT rather than the BS and BT groups. This would have homogenized the observed range shift rates between burned and unburned groups and decreased the probability of detecting an impact of fire on range shift rate. This implies that, with a more comprehensive ability to identify burned plots, the effects of fire on range shifts would have been even larger."

Also, upon reviewing the FIA documentation we realized that they stipulate that all fires recorded must have exhibited mortality *and/or damage* to 25% of trees. We have corrected this mistake in the text. (Line 323)

Range shifts: The study does not directly measure range shifts, but rather infers that the species' range is shifting or will shift, because the climatic niche of adult trees differs from the niche of juveniles (not really seedlings...since all trees are above breast height). This space-for-time substitution is a major limitation of the work, IF it's going to focus so heavily on range shifts. The results here have implications for considering how tree species' ranges will change with climate change...but this study did not measure range shifts. Somewhat like competition is presented in this study, the text is written (from the title onward) as if change over time was measured. My comment here largely echoes the comment in previous review #3: "There is a major mismatch between language and methods." The response is that text was added to clarify that the study focuses on climate space vs. geographic space, which is accurate, but the edited text notes that "we use 'leading edge' to refer to the leading edge in climate space." This does not really solve the issue, in my opinion: "leading edge" has a history of usage in the literature that inherently implies space. Redefining this in the text, while technically correct, is not a reasonable solution, because it will inherently be confusing to readers, at best, and potentially misleading.

We've attempted to clarify our use of 'leading edge' by supplanting the term throughout the text with 'potential leading edge'. We define the potential leading edge as plots that

have the potential to become a true leading edge because only the seedlings of a species are present, which have the potential to reach reproductive maturity.

(Lines 96-99) “We defined the source population as composed of plots with both trees and seedlings present and the potential leading edge as composed of plots with only seedlings present (if these seedlings reached reproductive maturity they may have eventually become a true leading edge).”

While we are not measuring range shifts in units of geographic distance/time, our approach is closer to this traditional definition and faces fewer limitations than a space-for-time substitution. Rather, we use a climate-space for geographic-space substitution and an age-for-time substitution. We add explicit reference to this substitution in Lines 99-102 “In effect, we calculated range shift velocity using a combination of climate-space for geographic-space substitutions and an age-for-time substitutions.” We see the primary limitations of this approach as (1) the difficulty of geographically contextualizing the range shifts (and corresponding rates) observed in climate space and (2) the uncertainty that seedlings and juvenile trees will reach maturity. The 2nd limitation most threatens our proxy range shift metric, but as stated elsewhere, we attempt to mitigate this by working to identify and exclude species where observed range shifts were not consistent across multiple life stages (Lines 195-201)

2. The framing of this study in the introduction, and placing the results in the context of other work in the discussion misses some key literature on this topic. For example, I was quite surprised that previous work by Dobrowski et al. (2015), who used FIA data to examine the climatic niche of adult and juvenile conifers across the western US, was not cited in the paper. This seems like a key foundation, conceptually and methodologically, for the current paper. There is also a host of recent literature discussing the evidence and potential causal mechanisms for a lack of post-fire conifer regeneration – required for a range shift - and the potential for fire to catalyze changes in forest communities that would otherwise occur more slowly due to climate change alone. Some of this work, which includes individual studies, larger syntheses, and topical reviews, seems at least potentially relevant to the framing of this study and the interpretation of the results: (Donato et al. 2016, Harvey et al. 2016, Johnstone et al. 2016, Crausbay et al. 2017, Davis et al. 2018, Stevens-Rumann et al. 2018, Parks et al. 2019, Turner et al. 2019, Coop et al. 2020). And, a new paper recently published by McLauchlan et al. (2020) provides potentially useful background and context for a current understanding of fire as a fundamental ecological process.

We appreciate the recommended citations and have integrated them into the intro (Lines 71-74, 80, 86-88, 95) and discussion (222-236). Because we have backed-off from analyzing the specific mechanisms by which fire could have explained the difference in range shift rates, some of the citations did not quite fit.

Minor comments:

- L 58-60 – This comes off as odd motivation for this work: wildfire as “a promising avenue for studying the effects of competition on range shifts”? This speaks a bit to the forcing of this work into the range-shift topic, vs. approaching this as a question about how wildfire may or may not alter range shifts expected due to climate change alone. The latter framing seems more relevant for a broad-audience journal like Nature Communications.

We hope that the more concise sentence we replaced this with communicates our intentions more directly. This project was largely motivated with competition in mind, but this phrase you’ve cited fell short of accurately communicating our intention. More specifically we’re trying to falsify the hypothesis that competition affects range shifts and learn more about the effect of wildfire on range shifts at the same time (Now Lines 65-70).

- The use of acronyms seems unnecessary and made it more challenging for me to follow the text. For example, does “range shift” really require the acronym “RS”? In my experience, this type of writing makes the text less accessible to a broad audience.

We’ve expanded all RS to range shift in an effort to make this more clear.

- Much of the text at the front end of the discussion reads more like introductory text – e.g., L 179-198. It’s very broad -- setting up context and expectations or hypotheses motivating the work -- vs. placing results in the context of existing literature and current understanding.

Reviewer 3 expressed similar opinions on the first few paragraphs of the discussion section and we have now moved much of the material (that wasn’t redundant) into the ‘main’ section of the manuscript. We’ve also attempted to further contextualize the work in existing literature.

- The consistent use of verbs like “changed” “decreased” “increased” imply change over time, when in fact these differences are across space, over essentially one point in time. For example, in the introduction (L 73): “...we used observational evidence...to test the hypothesis that...fires facilitated plant range shifts in response to...climate change.” This is in part what make the reader think that “range shifts” (in space, over time) are being measured, when in fact, this is not the case. As noted above, the results have implications for considering change over time, but they do not provide direct evidence of change over time. The results have implications for understanding change over time, with the important assumptions of space-for-time substitutions, and that would ideally be clear in the discussion.

We understand how this may be confusing and have implemented some changes to clear this up. We have been more explicit about the way we approximate range shift rate following the sentence (what was line 73) that you cited and have added a more explicit description of the age-for-time substitution that we implemented.

(Lines 94-102) “We analyzed the difference in the climatic niches of mature and juvenile trees, much like Dobrowski et al. 2015, and used these differences in climate-space to approximate shifts in geographic space. We defined the source population as composed of plots with both trees and seedlings present and the potential leading edge as composed of plots with only seedlings present (if these seedlings reached reproductive maturity they may have eventually become a true leading edge). In effect, we calculated range shift velocity using a combination of climate-space for geographic-space substitutions and age-for-time substitutions (unless otherwise specified, “distance” refers to climate distance and implied changes over time refer to differences between age groups throughout the text).”

We initially approached this comment by going back through the results and changing all ‘increased’ ‘decreased’ terms to time-neutral terms, but found that this made the text much more abstruse in some places.

References from this review (with * indicating review pieces)

*Coop, J. D., S. A. Parks, C. S. Stevens-Rumann, S. D. Crausbay, P. E. Higuera, M. D. Hurteau, A. Tepley, E. Whitman, T. Assal, B. M. Collins, K. T. Davis, S. Dobrowski, D. A. Falk, P. J. Fornwalt, P. Z. Fulé, B. J. Harvey, V. R. Kane, C. E. Littlefield, E. Q. Margolis, M. North, M.-A. Parisien, S. Prichard, and K. C. Rodman. 2020. Wildfire-Driven Forest Conversion in Western North American Landscapes. *Bioscience* 70:659-673.

Crausbay, S. D., P. E. Higuera, D. G. Sprugel, and L. B. Brubaker. 2017. Fire catalyzed rapid ecological change in lowland coniferous forests of the Pacific Northwest over the past 14,000 yr. *Ecology* 98:2356-2369.

*Davis, K. T., P. E. Higuera, and A. Sala. 2018. Anticipating fire-mediated impacts of climate change using a demographic framework. *Functional Ecology* 32:1729-1745.

Dobrowski, S. Z., A. K. Swanson, J. T. Abatzoglou, Z. A. Holden, H. D. Safford, M. K. Schwartz, and D. G. Gavin. 2015. Forest structure and species traits mediate projected recruitment declines in western US tree species. *Global Ecology and Biogeography* 24:917-927.

Donato, D. C., B. J. Harvey, and M. G. Turner. 2016. Regeneration of montane forests 24 years after the 1988 Yellowstone fires: A fire-catalyzed shift in lower treelines? *Ecosphere* 7.

Harvey, B. J., D. C. Donato, and M. G. Turner. 2016. High and dry: post-fire tree seedling establishment in subalpine forests decreases with post-fire drought and large stand-replacing burn patches. *Global Ecology and Biogeography* 25:655-669.

*Johnstone, J. F., C. D. Allen, J. F. Franklin, L. E. Frelich, B. J. Harvey, P. E. Higuera, M. C. Mack, R. K. Meentemeyer, M. R. Metz, G. L. W. Perry, T. Schoennagel, and M. G. Turner. 2016. Changing disturbance regimes, ecological memory, and forest resilience. *Frontiers in Ecology and the Environment* 14:369-378.

*McLauchlan, K. K., P. E. Higuera, J. Miesel, B. M. Rogers, J. Schweitzer, J. K. Shuman, A. J. Tepley, J. M. Varner, T. T. Veblen, S. A. Adalsteinsson, J. K. Balch, P. Baker, E. Batllori, E. Bigio, P. Brando, M. Cattau, M. L. Chipman, J. Coen, R. Crandall, L. Daniels, N. Enright, W. S. Gross, B. J. Harvey, J. A. Hatten, S. Hermann, R. E. Hewitt, L. N. Kobziar, J. B. Landesmann, M. M. Loranty, S. Y. Maezumi, L. Mearns, M. Moritz, J. A. Myers, J. G. Pausas, A. F. A. Pellegrini, W. J. Platt, J. Roozeboom, H. Safford, F. Santos, R. M. Scheller, R. L. Sherriff, K. G. Smith, M. D. Smith, and A. C. Watts. 2020. Fire as a fundamental ecological process: Research advances and frontiers. *Journal of Ecology* 108:2047-2069.

Parks, S. A., S. Z. Dobrowski, J. D. Shaw, and C. Miller. 2019. Living on the edge: trailing edge forests at risk of fire-facilitated conversion to non-forest. *Ecosphere* 10:e02651.

Parks, S. A., L. M. Holsinger, M. A. Voss, R. A. Loehman, and N. P. Robinson. 2018. Mean Composite Fire Severity Metrics Computed with Google Earth Engine Offer Improved Accuracy and Expanded Mapping Potential. *Remote Sensing* 10:879.

Stevens-Rumann, C. S., K. B. Kemp, P. E. Higuera, B. J. Harvey, M. T. Rother, D. C. Donato, P. Morgan, and T. T. Veblen. 2018. Evidence for declining forest resilience to wildfires under climate change. *Ecology Letters* 21:243-252.

Tepley, A. J., J. R. Thompson, H. E. Epstein, and K. J. Anderson-Teixeira. 2017. Vulnerability to forest loss through altered postfire recovery dynamics in a warming climate in the Klamath Mountains. *Global Change Biology* 23:4117-4132.

Turner, M. G., K. H. Braziunas, W. D. Hansen, and B. J. Harvey. 2019. Short-interval severe fire erodes the resilience of subalpine lodgepole pine forests. *Proceedings of the National Academy of Sciences*:201902841.

Reviewer 3

As far as I know, this is the first empirical test of the hypothesis proposed (which was suggested earlier by Pausa & Keeley 2019). Despite some methodological shortcomings (mostly recognised by the authors), the results are interesting and the manuscript could make a significant contribution to the topic. However, in the current version, some of the statistics and results are unclear, and the text needs to be much improved. The Discussion section is too long (some parts can be condensed, others moved to Methods); and tables and figures are poor (in the presentation and in the details of the caption). Below I provide some specific comments.

Title: Remove 'competition and' as you are not measuring competition; competition is inferred as possible mechanism.

We've added additional argumentation to the manuscript to defend our position that this study does provide some insight into the relationship between range shift rates and competition. Our argument is roughly that wildfire always leads to an immediate reduction in competition, and that the observed range shift rates were due, at least in part, to that reduction in competition. Our study cannot parse the various drivers of the difference in range shift rates we observe (certainly fire-adaptations affected the range shift rate of some species far more than reduction in competition), but because the reduction in competition almost certainly had some measure of impact we cannot refute the hypothesis that competition can affect range shift rates.

(Lines 67-70) "While the presence of an effect of fire in accelerating range shifts does not definitively prove a role for competition (the effect of fire on community assembly certainly

goes beyond the reduction competition), the absence of an effect of fire would provide strong grounds for rejecting the hypothesis.”

(Lines 202-206) “Fire adaptation and succession dynamics could confound the results where rates of range shifts into burned areas were entirely controlled by post-fire colonization adaptations, with no evidence of climate change as a driver. We minimized this effect by excluding species where the direction of range shift in burned areas was inconsistent with the direction of range shift in unburned areas.”

(Lines 210-211) “The nature of the FIA data precludes a deep exploration of the relative contributions of fire’s direct, abiotic impacts and its climate-change mediated, biotic impacts.”

That being said, we have change the title somewhat to put more stress on wildfire, because this is the driver that was directly studied.

Line 26, etc. - but see papers by W. Bond (e.g., 2005 in *New Phytol*, 2020 in *Trends Plant Sci*, etc.), where he and col. show that plant biogeography is not only determined by climate (your paper is another example).

Added “largely determined by climate” and cited Bond, 2020. We would note that the alternative stable states are still constrained by climate regimes.

Lines 60-61, 72-73, 164, ... You should mention that this hypothesis was proposed earlier (Pausas & Keeley *Front. Ecol & Environ.*), and your objective is to test it.

I’ve included this citation with other literature that proposed this through either theory or modeling work (Line 71).

Line 78: assuming that the presence of seedlings is a “leading edge” is a weak point.

We have changed “leading edge” to “potential leading edge” throughout the text

Table 1; how you (or FIA) differentiate between oak seedlings and oak resprouts?

FIA does not distinguish between seedlings and resprouts. They only tally resprouts that grow from a dead tree, and tall multiple resprouts as one seedling. We’ve added this to lines 328-330 in the methods. “FIA data do not distinguish between seedlings and resprouts and tally multiple suckers from the same root system (must be from a dead tree) as one seedling.” And we’ve added discussion of the implication of this for particular species that are known to resprout in the discussion (Lines 222-236)

Lines 117-122. Where are these results? I do not see DC and DCb/DCn values in the tables and figures.

DCb/DCn is simply the 3rd column of Table 2 divided by the 4th column. To make this clear we added a 5th column to Table 2 listing all of these values

There are terminological problems that makes the text confusing; e.g., the “treatments” are termed in different way in the text (Burned/Unburned), in the tables (Inside/Outside fire), and in the acronyms (B, N). Are the “Novel seedlings” in Table 1 the “Seedlings only” in the text?

What do 'NJ' and 'NG' mean in Table 1? Please take care and be consistent with the terminology used through the text, and include the meaning of the acronyms in tables (so they can be understood by their own).

We found most of our terminological inconsistencies between the text and Tables 1 and 2. We have updated these figures so that the terminology is consistent.

Table 2. * indicates significance of what? Different from 0? comparing Inside vs Outside?
In the caption of Table 2 we clarified that * indicates the statistically significant difference between potential leading edge and source population climatic niches. The null hypothesis for the niche equivalency test is that Schoener's $D = 1$ and the null hypothesis for the Hotelling's T^2 Test is that Centroid Distance = 0.

Fig. 3a. Please add the statistical significance between Burned and Unburned plots for each variable. I suppose that Fig. 3b only makes sense in the case that the difference between B and U (in 3a) is significant.

We tested the significance of the differences using a Wilcoxon Rank-Sum test and found that the differences for each climate variable are significant. We've added the p-values to the figure.

Lines 132-133: The values 100% and 208% are a bit surprising, perhaps because they are in %. It should be better explained. If these are important values, then a table (in the Suppl. Mat.) for each species and variable (with more details on how they were computed) would help.

We attempted to clear up this confusion by using "X times greater/less than" terminology when comparing values throughout much of the text, e.g. "The difference between range shift rates is greatest for Mean Winter Precipitation (MWP), where the average range shift rate is 3.4x greater in burned plots. (Lines 155-157)".

The values now in lines 152-158 are slightly different than those presented before because of the discovery of a rounding error in our code. We also added a more thorough derivation of these values to Appendix 1.

Line 136: 'range of RS' perhaps 'variability of RS'?
Implemented

Line 142: 'faster' is this significant?
As noted above, we performed a Wilcoxon Rank-Sum test and found this difference to be significant. (p-values were added to Figure 3a.)

Line 164: add Pausa & Keeley (above).
Added

Line 171: well, there are many examples of natural selection after 1 or 2 generations; the examples by Grant & Grant (beak sizes of finches) are a classical example, but there are many others.

We didn't mean to imply that natural selection cannot occur over 1 or 2 generations, but rather that rate of adaptation is unlikely to be fast enough to keep up with climate change.

We've added reference to Anderson & Wagymar 2020, which supports this claim. (Now Lines 46-49)

Line 182-184: the concept of "climax" is a bit outdated now, especially in a paper that have a dynamic view of nature as this one. See papers by Bond (e.g., Trends in Pl Sci) on state-shifts. **We changed the terminology to more clearly articulate our understanding of plant succession and state shifts without implying the Clementsian 'Climax Community' paradigm.**

(Lines 75-80) "Wildfire typically facilitates shifts in species composition along trajectories of ecological succession, where species that become abundant shortly after fire occurrence are eventually replaced. Post-fire succession can be predictable and specific to ecosystems and return an ecosystem to the initial pre-fire species assemblage. In contrast, fire can also facilitate state-shifts where the post-fire trajectory of vegetation succession leads to a different suite of species."

Line 188, 235-253; these two species have been studied in detail by Rodman et al. (2020 J Ecol.) in relation wildfire responses; how your results fit with those in that paper?

Because FIA fire data does not provide enough information to make specific claims about how different fire adaptation affected our results, we only discuss how some adaptations had the potential to affect the observed range shift rates. We've included a few sentences on *P. ponderosa* and *P. menziesii* (as well as others) in the discussion (Lines 222-236)

Lines 193-201. Could be condensed.

Much of this material was removed when this 'background info' on fire was merged into the introduction

Line 200: by ground fires do you mean surface fires? Understory fires?

FIA only provides a crown/ground fire dichotomy. We assume that surface fires are also classified as ground fires, and have added this consideration to (Lines 215-216).

Lines 203-4: High intensity (crown) fires are not always as negative as it seems here (sterilization, erosion, ...); it is likely a natural fire regime in the American boreal forest (Archibald et al. Envir. Res. Lett.). In fact some of your species may benefit from high intensity fires (is your *Pinus contorta* serotinous?).

We have abandoned our initial and simplistic characterization of 'low severity' = 'higher range shift rate' and 'high severity' = 'lower range shift rate'. The FIA data on fires does not provide enough information to produce meaningful analysis of how different types of fires affect range shift rate. In our discussion of fire in the discussion section, we've changed our approach to discuss variables of fire impacts that *could* have affected our observed range shift rates (Lines 210-236)

Line 210: again, is any of your species serotinous? He et al. New Phytol may help. And do they resprout? Probably *Populus tremuloides* and *Quercus chrysolepis* are good resprouters. How this affects your results?

We looked at He et al., 2012 and the USDA "Fire Effects Information System" (<https://www.feis-crs.org/feis/>) and determined only *Pinus contorta* subsp. *latifolia* is

serotinous (which is only present in the rocky mountain region of this study, whereas the non-serotinous *Pinus contorta* subsp. *murrayana* covers the non-rocky mountain regions). In addition to *P. tremuloides* and *Q. chrysolepis*, we found that *Lithocarpus densiflorus* can also resprout from burls following a fire. We've stated as much in the discussion and have added our thoughts on how this might affect our results (Lines 222-236)

Line 213: "locales" → "microsite"?

Implemented

Lines 235-245: see Rodman et al. (2020 J Ecol.)

Added this to Lines 233-236

261-265, methods

Removed most of the content of these sentences because this information is already present in the methods

263-267: to add the info in the Suppl Mat.

The information on the vector agreement between different age groups can be found in Table 1. I added a reference to Table 1 in the sentence you indicated (Lines 196-199) "To minimize the impact of sink populations on our results, we excluded species where the direction of range shift was not consistent with the observed range shift direction observed between saplings (2.5cm < dbh < 12.7cm) and large trees (dbh > 12.7cm) (Table 1)."

Line 264: "eight trees" → "eight species"? or "eight tree species"?

Implemented

The Discussion needs to be shortened and written in a more condensed way.

We've removed much of the text within the first few paragraphs of the discussion that was redundant information from the 'main' section.

I would say that Fig. 5 is not mentioned in the main text, only in the methods, if so, it would be better to move this figure to the Suppl. Mat.

We think it might be worth keeping Fig. 5 in the main text because it is important for understanding some of the core analysis of this work, and may be particularly helpful for a broader audience where vector math may not be intuitive

Fig. A1, the symbols differ between the plot and the legend.

Fixed the legend

Reviewer comments, second round review:

Reviewer #2 (Remarks to the Author):

See attached document.

Reviewer #3 (Remarks to the Author):

The manuscript has been improved from previous version, but it still has many things unclear that require clarification.

Title: Remove the term “competition”. Your study may provide some insights into the mechanism, but you are not measuring competition; competition is inferred as possible mechanism. (I mentioned this in my previous report).

Line 66: the same here, you are not testing competition; you are testing the role of fire, and changes in competition are a possible mechanism (the effect of fires are much more complex). You would need an experimental approach to proof that it is a direct effect of competition and it is not driven by another mechanism (e.g., removal of seed predators, release of allelopathies, removal of pests, etc...).

Lines 79-80: Would ref 3 (Pansas & Bond) be appropriate here?

Lines 86-88: Would Enright et al. (2015, *Frontiers EE* 2015) be appropriate here?

Line 11: In tables and figures you are using the term “Unburned”; would it be easier for the reader to use U instead of N?

Line 116 and Table 1. Which of these species resprout after fire? The results for those species are less robust (as you did not differentiate between seedlings and resprouts). This shortcoming needs to be addressed (I already mentioned this in my previous report).

Table 2, What is the point of computing the ration of the Centroid Distances (CD) when one of the Centroid Distances is not significant? E.g., in *Pseudotsuga* there is no climatic differences between Seedlings vs Seedlings+Trees (in burned plots) suggesting that there is no shift. But the CD rate suggests that in burned conditions the shifts is 2.15 times greater than in unburned conditions. Obviously something is wrong. The same happens in other species.

Line 143: Does *Lithocarpus* resprout after fire? If so, how can this affects this results?

Fig. 1. Unclear, I cannot see the blue symbols (plots) in the map.

Fig. 2. Please make the figure squared! i.e., with a similar length for one unit in x and y axes. This is for one species, the corresponding figures for the others species should be in the Online Suppl information.

Fig. 3a. It is hard to believe that the burned and unburned values are significantly different given that boxplots overlap so much. Are you really using the correct stats?
How this is done? This include all the species? If so, then the species should be a random factor, I guess. Please provide details and make sure you are using robust stats, which doesn't seems the case now.

Fig. 3a,b. I find the legend confusing: "RS into unburned"? - I suppose that RS means 'range shift' but it is not mentioned in the caption (so please make sure that the figures can be understood easily). But also it is unclear the "into unburned". I understood that the shift is within unburned not from burned into unburned.

Lines 640: what is a z-transformation? I think it is not mentioned in the methods.

Lines 642,643: "shows the value of range shift rate relative to unburned plots", should it be "... of burned plots relative to unburned plots"?

Fig. 5 could be moved to the Online supp mat

lines 191-194: what about the seedling-resprouting problem?

192 by 'sink population' are you referring to the sapling bank?

219: can ... but not necessarily!

226: Unclear. If they resprout, it is impossible to have 100% mortality.

312: not tested – you tested a pattern not a mechanism.

Review of revision of Avery and Field, “Wildfire, Competition and the Rate of Tree Range Shifts in the American West”

Philip Higuera, University of Montana, 20 Feb., 2021.

The revised manuscript is significantly improved, addressing many of the comments and critiques raised in previous reviews. I appreciate the scope of this revision and believe the impact of the work will be improved because of these efforts. There are a few key comments in the detailed comments below that, if addressed, would clarify the logic of the study, improve accuracy of species interpretations, and/or make the study understandable to a broader audience:

[a] Logic on how competition is accounted for is much improved. Nonetheless, I still found myself hung up on what was considered competition in the logic set out in the new text in the introduction (L 64-79). Minor edits could address this comment.

[b] Text and description of range of lodgepole pine and three subspecies (L 228) should be updated to improve accuracy. Text is misleading/incomplete as written.

[c] Explanation for odd/unexpected pattern in lodgepole pine and subalpine fir could have more clarity if text was edited/revised (c. L 247).

[d] Text on “high altitude” forests, like the incomplete description of lodgepole pine, adds confusion to the text and/or lacks accuracy (L 277-285).

[e] Finally, it should be pointed out in the method why “American West” excludes forests across much of the West (WY, CO, UT, NM, AZ).

Detailed comments

L 22: Suggest “...and that *fire and* fire management may play a role..” OR “...and that *fire and* fire management *will* play a role...” The emphasis on the role of fire management is important; but/and, fire will play a role, regardless of fire management.

L 65-79: In this new text, aimed at clarifying how the study is specifically getting at competition, the logic still seems a little forced. For example, in a world with no interspecific competition, fire could still accelerate range shifts, via mortality (i.e., killing mature trees, which germinated when climate was different). OR...is the paper also considering intraspecific competition, such that increased substrate, light, water, etc., from adult mortality qualifies as reduced competition? Writing that out here, it makes sense, but it might be helpful to explicitly note that the rationale is that any tree mortality necessarily results in reduced competition, if that is indeed the logic.

L 181: This pattern with subalpine fir is odd...and I was glad to see it addressed more thoroughly later in the text. It remains a challenging pattern to explain, given that this species generally grows in cool/cold, moist climates.

L 207: Strongly suggest not using “normal colonization” here. For some of these species, particularly subalpine species (subalpine fir and lodgepole pine), post-fire regeneration accounts for regeneration over most forested areas. That is, forest age structures generally reflect the

history of high-severity fire. This could be avoided by just writing "...different from colonization in the absence of fire."

L 215-221: This new text helps clarify the nature of the dataset nicely.

L 216: Consider: "...the "ground fire" categorization reflectd surface fire. *Does FIA just uses the term "ground" instead of "surface"?"

L 221: "...response to the varied impacts *of* wildfire."

L 226: Using "designed" here opens up an entirely different field of inquiry. Suggest: "The serotinous cones (which open and release seeds after being heated by fire) of..."

L 228: By calling out var. *murrayana*, not referencing var. *contorta*, and seemingly minimizing the extent of var. *latifolia* without any quantitative information, this text on subspecies of lodgepole pine comes off as slightly naive of its distribution and ecology. Some revision would improve the paper. The distribution of Rocky Mountain lodgepole pine (var. *latifolia*), potentially with serotinous cones, includes central and eastern OR and WA, east of the Cascades, in addition to dominating many forests in MT and ID:

<https://www.fs.fed.us/database/feis/plants/tree/pinconl/all.html>,

https://www.conifers.org/pi/Pinus_contorta_latifolia.php. The study region (Fig. 1) also includes the range of var. *contorta*, in WA and along the coast of WA, OR, and CA; if lodgepole is not included in the FIA plots in these area, that should be pointed out:

https://www.conifers.org/pi/Pinus_contorta.php.

L 231: This point on fire-adapted traits is a key limitation or caveat, particularly in forests that burn infrequently (e.g. subalpine forests). This may help explain the odd pattern for subalpine fire and lodgepole pine, somewhat. In forests that historically burned at high severity, at intervals of centuries or more, subalpine fir would be replaced seral species, but would/could be present in greater abundance in recently burned stands. The same could apply to lodgepole pine, where it's replaced by Engelmann spruce (and subalpine fir) with greater time-since-fire.

L 235: Suggest "its longer dispersal distance." vs. the value-laden "better."

L 240-243: I found this new text helpful.

L 243: Suggest "could" here vs. "would," as the point is that you don't know, given the lack of information on fire occurrence prior to five yr before sampling. For example, if "burned" included sites that burned within the previous 10 or 20 yr – which surely some of the FIA sites fit -- wouldn't one expect the signal of burned sites to be diluted?

L 247-248: Suggest lessening the emphasis on "fire adaptation" (which is difficult to support, evolutionarily) and just highlight "...particular traits that aid in post-fire regeneration (like the aforementioned serotinous cones, or re-sprouting)..." While serotiny in lodgepole pine is one of the few traits that has support for selection by fire, re-sprouting is a common trait that can be selected for by many different factors, not necessarily including fire. It's a trait that happens to confer a competitive advantage in a post-fire environment. And...perhaps more importantly,

isn't it true that "population growth is accomplished by recolonization from nearby populations" for all species that are killed by fire (high severity or otherwise)?

L 249-251: The focus on canopy gaps and microsites here seems to suggest that fires are small-scale disturbances. High-severity patches within large wildfires, with near 100% tree mortality, can vastly exceed the scale of individual stands or microsites. Overall...it's possible that the text from "In general..." (L 245) to "...and species traits" (L 252) could be deleted. The next sentence – "We expect that..." is the key point here.

L 257: Suggestion: "...establishing in recently burned areas."

L 259: This study spanned many different ecosystems, so this text – "...on this and other ecosystems,..." does not make sense; it suggests this study focused on one ecosystem.

L 260: "...interactions between *post*-fire succession and..."

L 263: CMIP – many will know what this means, but likely many will not. Spell out?

L 277-285: This paragraph on subalpine fir and lodgepole pine is confusing and does not make sense based on the ecology and distribution of these species. Part of the issue may come from using altitude as a filter, vs. climate. Consider using "subalpine forests" as the descriptor here, vs. altitude: it's more accurate (and does not require these caveats). As the authors know, elevation is mainly important insofar as it affects climate, and across such a range of latitude and distance-from-ocean, the elevation of subalpine forests varies widely across the West. Subalpine fir and lodgepole pine occur in subalpine forests across the region. Lodgepole also occurs in some montane and lower-elevation forests, particularly var. *contorta*. The "typically relegated to high altitude" thus comes off as misleading, and it also suggests that subalpine forests don't cover broad areas.

L 278: The reference here to "southern Rocky Mountains" is confusing, as this often/usually refers to either CO/WY and/or NM/AZ – areas south of the "Northwestern Forested Mountains" ecoregion used in this study. This would only make sense if the Canadian Rockies were the northern Rockies, and it would still leave the Rockies through WY/CO/AZ/NM/UT unaccounted for. I recognize the ecoregion names are sometimes odd (e.g., some of your study area is considered "Canadian Rockies" in some schemes, even though in the USA). Also, as noted above, the text here on *P. contorta* comes off as not recognizing that this species dominates across the entire study region – not just the Sierra Nevada.

Finally, why does "the American West" exclude the central and southern Rocky Mountains and Great Basin? There are plenty of forests across these regions, although not as widespread as in the studied ecoregions. Some/many of the studies cited in the text come from the Rocky Mountains outside of the areas included here. The rationale for this exclusion is important to note in the methods. The title, as such, is slightly misleading, although I appreciate that one cannot just add more geographic specificity. Upon reading the revision, it took me a while to realize that "the American West" was excluding nearly 1/4-1/3 (?) of forested areas. If the title remains, it would be helpful to provide the rationale for the study region in the intro. text.

Line by Line Response

Reviewer 2:

The revised manuscript is significantly improved, addressing many of the comments and critiques raised in previous reviews. I appreciate the scope of this revision and believe the impact of the work will be improved because of these efforts. There are a few key comments in the detailed comments below that, if addressed, would clarify the logic of the study, improve accuracy of species interpretations, and/or make the study understandable to a broader audience:

[a] Logic on how competition is accounted for is much improved. Nonetheless, I still found myself hung up on what was considered competition in the logic set out in the new text in the introduction (L 64-79). Minor edits could address this comment.

We have edited the text here to be more specific about what we mean when using the term ‘competition’. (Lines 66-70)

“Wildfire, because it immediately reduces vegetation cover and therefore alters the availability of light and water, provides an opportunity to test the hypothesis that the relaxation of these resource constraints can affect range shift rates”

[b] Text and description of range of lodgepole pine and three subspecies (L 228) should be updated to improve accuracy. Text is misleading/incomplete as written.

We’ve added a more thorough description of each subspecies ranges to this section (Lines 239-246)

[c] Explanation for odd/unexpected pattern in lodgepole pine and subalpine fir could have more clarity if text was edited/revised (c. L 247).

Stimulated by your detailed comments on this point, we’ve updated the text to enhance clarity and added more context on the different scales at which this recolonization mechanic might operate (Lines 260-266)

“Following a fire, population establishment and growth is accomplished by recolonization from adjacent populations unless species have particular traits that aid in post-fire regeneration (like the aforementioned serotinous cones or re-sprouting). A gap in the canopy or a fire-cleared hillside potentially promotes establishment through reduced competition”

[d] Text on “high altitude” forests, like the incomplete description of lodgepole pine, adds confusion to the text and/or lacks accuracy (L 277-285).

We’ve reworked this section, replacing ‘high altitude’ forests with the more accurate ‘subalpine forest’. We’ve also revised our potential explanation of the subalpine species range shift to be clearer and to emphasize that this is a hypothesis and not something we tested in this study (Lines 292-298)

“The two species that did not have seedling-only plots biased towards greater MSP or lesser MTWM were *P. contorta* and *A. lasiocarpa*, the only two species in this study that populate subalpine regions. This could reflect the constraint that limited potential

establishment sites at the upper-elevation margins of their distributions leads to a greater abundance of seedling-only plots at the lower-elevation margins..."

[e] Finally, it should be pointed out in the method why "American West" excludes forests across much of the West (WY, CO, UT, NM, AZ).

During preliminary data analysis we excluded WY based on our understanding at the time that Wyoming's FIA data did not include the level of plot detail necessary for the analysis. This was a misunderstanding. It resulted in an exclusion of around ~3000 plots (less than 10% of the 33000 plots we used in this study). We've noted the omission of WY, as well as disjunct patches in CO, UT, NM, and AZ, in both the introduction and methods (Lines 94-95, 345-349):

"We sourced all plots (33,838 total) within the Northwestern Forested Mountains and Marine West Coast Forest ecoregions of the continental U.S. ⁶⁷ (Fig. 1), omitting FIA plots in Wyoming and in disjunct patches of forest in Utah, Colorado, and New Mexico. The omission of Wyoming was inadvertent and would have increased the sample size by less than 10%. The subsequent omission of plots in disjunct patches (CO, NM, UT) limited the risk of considering forest areas too small for meaningful range shifts. "

Detailed comments

L 22: Suggest "...and that fire and fire management may play a role.." OR "...and that fire and fire management will play a role..." The emphasis on the role of fire management is important; but/and, fire will play a role, regardless of fire management.

We agree and have implemented this suggestion (Lines 23)

L 65-79: In this new text, aimed at clarifying how the study is specifically getting at competition, the logic still seems a little forced. For example, in a world with no interspecific competition, fire could still accelerate range shifts, via mortality (i.e., killing mature trees, which germinated when climate was different). OR...is the paper also considering intraspecific competition, such that increased substrate, light, water, etc., from adult mortality qualifies as reduced competition? Writing that out here, it makes sense, but it might be helpful to explicitly note that the rationale is that any tree mortality necessarily results in reduced competition, if that is indeed the logic.

As discussed in our response to your first suggestion (point [a]), we revised this section to supplant the term 'competition' with a more exact description of the relationship between fire and range shift rates. (Lines 66-70)

L 207: Strongly suggest not using "normal colonization" here. For some of these species, particularly subalpine species (subalpine fire and lodgepole pine), post-fire regeneration accounts for regeneration over most forested areas. That is, forest age structures generally reflect the history of high-severity fire. This could be avoided by just writing "...different from colonization in the absence of fire."

Agreed and Implemented (Lines 215)

L 216: Consider: "...the "ground fire" categorization reflected surface fire. *Does FIA just uses the term "ground" instead of "surface"?

We changed this phrase to "'ground fire' categorization refers to surface fires" because it seems increasingly clear that these are synonyms. It does not seem reasonable that FIA would have recorded the presence of subterranean fires, which some refer to as 'ground fires'. (Lines 224)

L 221: "...response to the varied impacts of wildfire."

Added the 'of' as suggested (Lines 229)

L 226: Using "designed" here opens up an entirely different field of inquiry. Suggest: "The serotinous cones (which open and release seeds after being heated by fire) of..."

We've changed this text to your suggestion (Lines 239-240)

L 228: By calling out var. murrayana, not referencing var. contorta, and seemingly minimizing the extent of var. latifolia without any quantitative information, this text on subspecies of lodgepole pine comes off as slightly naive of its distribution and ecology. Some revision would improve the paper. The distribution of Rocky Mountain lodgepole pine (var. latifolia), potentially with serotinous cones, includes central and eastern OR and WA, east of the Cascades, in addition to dominating many forests in MT and ID:

<https://www.fs.fed.us/database/feis/plants/tree/pinconl/all.html> ,

https://www.conifers.org/pi/Pinus_contorta_latifolia.php . The study region (Fig. 1) also

includes the range of var. contorta, in WA and along the coast of WA, OR, and CA; if lodgepole is not included in the FIA plots in these area, that should be pointed out:

https://www.conifers.org/pi/Pinus_contorta.php.

As noted above in response to your suggestion [b] we've added some biogeographical context for each subspecies using your provided sources and have removed text that could have been interpreted as minimizing the extent *Pinus contorta* subsp. *latifolia*. (Lines 239-246)

L 231: This point on fire-adapted traits is a key limitation or caveat, particularly in forests that burn infrequently (e.g. subalpine forests). This may help explain the odd pattern for subalpine fire and lodgepole pine, somewhat. In forests that historically burned at high severity, at intervals of centuries or more, subalpine fir would be replaced by seral species, but would/could be present in greater abundance in recently burned stands. The same could apply to lodgepole pine, where it's replaced by Engelmann spruce (and subalpine fir) with greater time-since-fire.

We agree that it is important to think about seral replacements, but the essence of the pattern in the results is that, for both species, seedling only plots are in sites that are warmer and drier than tree plus seedling plots. The observation that the pattern is the same with and without fire doesn't suggest an explanation related to time since fire. We thought a lot about the unusual pattern for the two species and, as explained above, suspect that it is driven by the fact that, independent of range shifts, these two species have more suitable seedling habitat on the downslope than on the upslope ends of their distributions. The revised text

(Lines 292-298) reads, “The two species that did not have seedling-only plots biased towards greater MSP or lesser MTWM were *P. contorta* and *A. lasiocarpa*, the only two species in this study that populate subalpine regions. This could reflect the constraint that limited potential establishment sites at the upper-elevation margins of their distributions leads to a greater abundance of seedling-only plots at the lower-elevation margins...”

L 235: Suggest “its longer dispersal distance.” vs. the value-laden “better.” L 240-243: I found this new text helpful.

Implemented (Lines 252)

L 243: Suggest “could” here vs. “would,” as the point is that you don’t know, given the lack of information on fire occurrence prior to five yr before sampling. For example, if “burned” included sites that burned within the previous 10 or 20 yr – which surely some of the FIA sites fit -- wouldn’t one expect the signal of burned sites to be diluted?

Implemented to increase accuracy (Lines 259). We completely agree that the possibility of older fires in the sites listed as unburned by the FIA means that the difference we identify between burned sites and unburned sites is a minimum estimate for the difference between recently burned sites and sites not burned for many decades.

L 247-248: Suggest lessening the emphasis on “fire adaptation” (which is difficult to support, evolutionarily) and just highlight “...particular traits that aid in post-fire regeneration (like the aforementioned serotinous cones, or re-sprouting)...” While serotiny in lodgepole pine is one of the few traits that has support for selection by fire, re-sprouting is a common trait that can be selected for by many different factors, not necessarily including fire. It’s a trait that happens to confer a competitive advantage in a post-fire environment. And...perhaps more importantly, isn’t it true that “population growth is accomplished by recolonization from nearby populations” for all species that are killed by fire (high severity or otherwise)?

We’ve changed the noted text to “...particular traits that aid in post-fire regeneration” and have rearranged the sentence to more clearly communicate that we expect recolonization to occur from offsite populations unless their onsite populations have particular regeneration strategies. (Lines 262-265)

“Population establishment and growth is accomplished by recolonization from nearby populations unless species have particular traits that aid in post-fire regeneration (like the aforementioned serotinous cones or re-sprouting).”

L 249-251: The focus on canopy gaps and microsites here seems to suggest that fires are small-scale disturbances. High-severity patches within large wildfires, with near 100% tree mortality, can vastly exceed the scale of individual stands or microsites. Overall...it’s possible that the text from “In general...” (L 245) to “...and species traits” (L 252) could be deleted. The next sentence – “We expect that...” is the key point here.

We think that this section might be important for communicating the general mechanics of the phenomena that we otherwise discuss at a larger scale. We’ve added some text to

communicate that this recolonization happens at various scales (from microsite to much larger). (Lines 265-268)

e.g. “A gap in the canopy or a fire-cleared hillside potentially promotes establishment through reduced competition”

L 257: Suggestion: “...establishing in recently burned areas.”

Implemented (Lines 273)

L 259: This study spanned many different ecosystems, so this text – “...on this and other ecosystems,...” does not make sense; it suggests this study focused on one ecosystem.

Changed to “... on these and other ecosystems” (Line 275)

L 260: “...interactions between post-fire succession and...”

Implemented (Line 276)

L 263: CMIP – many will know what this means, but likely many will not. Spell out?

We’ve added the full name alongside the acronym (Line 279)

L 277-285: This paragraph on subalpine fir and lodgepole pine is confusing and does not make sense based on the ecology and distribution of these species. Part of the issue may come from using altitude as a filter, vs. climate. Consider using “subalpine forests” as the descriptor here, vs. altitude: it’s more accurate (and does not require these caveats). As the authors know, elevation is mainly important insofar as it affects climate, and across such a range of latitude and distance-from-ocean, the elevation of subalpine forests varies widely across the West. Subalpine fir and lodgepole pine occur in subalpine forests across the region. Lodgepole also occurs in some montane and lower-elevation forests, particularly var. *contorta*. The “typically relegated to high altitude” thus comes off as misleading, and it also suggests that subalpine forests don’t cover broad areas.

We’ve reworked much of this section as noted elsewhere, and have replaced “high altitude” with “subalpine”. (Lines 292-298)

L 278: The reference here to “southern Rocky Mountains” is confusing, as this often/usually refers to either CO/WY and/or NM/AZ – areas south of the “Northwestern Forested Mountains” ecoregion used in this study. This would only make sense if the Canadian Rockies were the northern Rockies, and it would still leave the Rockies through WY/CO/AZ/NM/UT unaccounted for. I recognize the ecoregion names are sometimes odd (e.g., some of your study area is considered “Canadian Rockies” in some schemes, even though in the USA). Also, as noted above, the text here on *P. contorta* comes off as not recognizing that this species dominates across the entire study region – not just the Sierra Nevada.

After a more careful look at the elevation profile of these species within our study area, it doesn’t make much sense to describe a Southern/Northern Rocky Mountain dichotomy. Upon the omission of this dichotomy the reader shouldn’t be led to believe that *P. contorta* et al. aren’t distributed across much of the study area. (Lines 292-298)

Finally, why does “the American West” exclude the central and southern Rocky Mountains and Great Basin? There are plenty of forests across these regions, although not as widespread as in the studied ecoregions. Some/many of the studies cited in the text come from the Rocky Mountains outside of the areas included here. The rationale for this exclusion is important to note in the methods. The title, as such, is slightly misleading, although I appreciate that one cannot just add more geographic specificity. Upon reading the revision, it took me a while to realize that “the American West” was excluding nearly 1/4-1/3 (?) of forested areas. If the title remains, it would be helpful to provide the rationale for the study region in the intro. text.

We provide a more detailed response to this point earlier, but you can find that we added a more accurate description of the study area in both the intro and methods. (Lines 94-95, 345-349)

Reviewer 3:

The manuscript has been improved from previous version, but it still has many things unclear that require clarification.

Title: Remove the term “competition”. Your study may provide some insights into the mechanism, but you are not measuring competition; competition is inferred as possible mechanism. (I mentioned this in my previous report).

Changed title to *Wildfire and the Rate of Tree Range Shifts in the American West*

Line 66: the same here, you are not testing competition; you are testing the role of fire, and changes in competition are a possible mechanism (the effect of fires are much more complex). You would need an experimental approach to prove that it is a direct effect of competition and it is not driven by another mechanism (e.g., removal of seed predators, release of allelopathies, removal of pests, etc...).

Reviewer 2 shared similar concerns with this section, and we’ve supplanted the term “competition” with a more specific description of how we think fire occurrence affects range shifts. (Lines 66-70)

“Wildfire, because it immediately reduces vegetation cover and therefore alters the availability of light and water, provides an opportunity to test the hypothesis that the relaxation of these resource constraints can affect range shift rates”

Lines 79-80: Would ref 3 (Pansas & Bond) be appropriate here?

We’ve added the reference here. (Lines 82)

Lines 86-88: Would Enright et al. (2015, *Frontiers EE* 2015) be appropriate here?

We’ve added this citation (Line 90)

Line 11: In tables and figures you are using the term “Unburned”; would it be easier for the reader to use U instead of N?

We've replaced the Ns with Us in acronyms throughout the text and figures

Line 116 and Table 1. Which of these species resprout after fire? The results for those species are less robust (as you did not differentiate between seedlings and resprouts). This shortcoming needs to be addressed (I already mentioned this in my previous report).

We listed the species that resprout in lines 231-233. We have further explained our interpretation of the role of resprouting in Lines 233-239. Essentially, a resprout will not show up in our potential leading-edge plots because living *and* dead mature trees are tallied by the FIA, and it would be unlikely that the tree from which a resprout was growing from would be unrecognizable/untallied.

"FIA data do not distinguish between resprouts and seedlings, and some resprouts may have been misclassified as seedlings in our analysis. However, this would be unlikely to inflate the calculated range shift rate of these species because resprouts would never satisfy the potential leading-edge classification: the presence of seedlings and absence of living or dead mature trees. Resprouting would inflate the range shift rate of a species in the unlikely case that a plot contained only resprouts of that species and the mature trees were burned beyond species-level identification."

Table 2, What is the point of computing the ration of the Centroid Distances (CD) when one of the Centroid Distances is not significant? E.g., in *Pseudotsuga* there is no climatic differences between Seedings vs Seedlings+Trees (in burned plots) suggesting that there is no shift. But the CD rate suggests that in burned conditions the shifts is 2.15 times greater than in unburned conditions. Obviously something is wrong. The same happens in other species.

The sample size is small for the burned seedling-only sites and the Hotelling's T^2 test of significant difference between centroids is sensitive to this small sample size. We think the DC_B/DC_U values may be worth reporting even without statistical significance, but we have gone back through the text (Lines 146-153) and the caption of Table 2 and carefully ensured that it's clear that DC_B/DC_U is reliable only for *Pinus ponderosa* and *Lithocarpus densiflorus*. "However, DC_B/DC_U is reliable only for *Pinus ponderosa* and *Lithocarpus densiflorus* because the DC_B of most species is not statistically significant (Hotelling's T^2 test, $p < .05$), likely owing to the small sample size of burned plots per species. The average DC_B/DC_U across all eight species is 2.10 (5.01 for the two statistically significant species)."

Line 143: Does *Lithocarpus* resprout after fire? If so, how can this affects this results?

***L. densiflorus* does resprout after fire and we've made this more explicit (Lines 231-233). Our response to your feedback at Line 116 above explains why resprouting does not get interpreted as a range shift, but the relevant text in the manuscript is at Lines 233-239**

Fig. 1. Unclear, I cannot see the blue symbols (plots) in the map.

We've remade the figure to not only increase legend clarity but also more effectively show plot density across the study area

Fig. 2. Please make the figure squared! i.e., with a similar length for one unit in x and y axes.

This is for one species, the corresponding figures for the others species should be in the Online Suppl information.

We changed the axes so that Figure 2 has the same scale for the x and y axes. We've also added an Appendix 7 where the PCA plots for all other species can be found.

Fig. 3a. It is hard to believe that the burned and unburned values are significantly different given that boxplots overlap so much. Are you really using the correct stats? How this is done? This include all the species? If so, then the species should be a random factor, I guess. Please provide details and make sure you are using robust stats, which doesn't seems the case now.

Upon revisiting the statistics of this figure we decided to overhaul the analysis and presentation of these data to make it simultaneously more clear and statistically sound. Instead of plotting the raw values of the differences in climatic niches (Potential Leading Edge – Source Population), we show the climatic niches of both Potential Leading Edge and Source Population and the linear regression between them. Additionally, we exclude species where the shift in climate space for that particular variable was not statistically significant. We found that this new figure summarizes the patterns and their statistical significance in a more clear and convincing way.

A detailed description of the production of this figure is now in the caption.

Fig. 3a,b. I find the legend confusing: “RS into unburned”? - I suppose that RS means ‘range shift’ but it is not mentioned in the caption (so please make sure that the figures can be understood easily). But also it is unclear the “into unburned”. I understood that the shift is within unburned not from burned into unburned.

We've entirely reworked Figure 3 and have been more explicit in our new legend.

Lines 640: what is a z-transformation? I think it is not mentioned in the methods.

This is the default standardization method of R's base function scale(). We've replaced “z-tranformation” with the more commonly used “Z-Score standardization” and also added a brief explanation in Figure 3s caption.

Lines 642,643: “shows the value of range shift rate relative to unburned plots”, should it be “... of burned plots relative to unburned plots”?

Upon reworking Figure 3 this is no longer relevant.

Fig. 5 could be moved to the Online supp mat

Based on explaining the study to several audiences, we find that this figure is important for helping even experts understand the fundamentals of our analysis, and could be critical in making this study accessible to the broad audience of this journal

lines 191-194: what about the seedling-resprouting problem?

As explained above, we now further explain our interpretation of the role of resprouting in Lines 233-239. Essentially, a resprout will not show up in our potential leading-edge plots

because living *and* dead mature trees are tallied by the FIA, and it would be unlikely that the tree from which a resprout was growing from would be unrecognizable/untallied.

192 by 'sink population' are you referring to the sapling bank?

We are referring to populations where the seedlings may never reach maturity. To make this point clearer, we replaced "sink" with "non-maturing" (Lines 200-211, 390)

219: can ... but not necessarily!

Changed to "Higher intensity fires (of which many are crown fires) can, but not necessarily, cause soil sterilization, nutrient depletion, and greater erosion" (Lines 227)

226: Unclear. If they resprout, it is impossible to have 100% mortality.

We have reworked this section entirely to be add clarity to this point. Namely, we've described how resprouts shouldn't affect range shift rate because plots with only resprouts will never be classified as "seedling only"/"potential leading-edge" plots, due to the fact that FIA data ID both living and dead trees, and any resprouts would be part of the "source population" plot category (Lines 231-239)

312: not tested – you tested a pattern not a mechanism.

We reworded this section to more explicitly identify this as a pattern not a mechanism (Lines 323-325)

"this pattern supports the argument that range shift rates are likely affected by reductions in population size and density of competitors due to wildfire."

Reviewer comments, third round review:

Reviewer #2 (Remarks to the Author):

See attached review.

Reviewer #3 (Remarks to the Author):

The manuscript tests a nice idea that is worth persuading. The results show a clear pattern supporting the hypothesis at least for one species (of the 12 species with data); another species also show the pattern but the authors recognise the small sample size; three additional species partially show the pattern, one of them include serotinous populations and thus their range shift may be inflated (line 241); and some species seems to shift in the opposite direction, i.e., to warmer sites (line 293-...). So I feel that there is an interesting pattern although I must recognise that the pattern is not very strong.

The writing still needs improving, it should be written with more precision; the quality of the figures is also limited.

Specific comments by line number

20: For the abstract, the number of species considered is more important than the number of plots.

45: "(1) the environment or disturbance" (typically disturbance is not considered an environmental variable)

71: The absence of effect cannot reject an (alternative) hypothesis!

84-86: sentence unclear

86: "south" – are you still talking about Alaska??

Fig. 1: it is not easy to differentiate the size of the symbols (number of plots) in the map.

118-119: excluded? Because it doesn't support your hypothesis? Perhaps the species that do not behave as hypothesised (4 of 12) should not be removed from Table 1, they are part of the results.

Also in Table 1 should mark the species that are resprouters or serotinous, e.g., with an (R) or (S) after the name. And mention that the number of seedlings of these species are likely to be overestimated.

124, 125: repetition (0.82-0.56 vs less than 0.82)

113-136: The first section of the results ("Evidence of range shifts in unburned plots") doesn't seem to respond to your hypothesis; removing it doesn't seem to affect the paper. Results (in relation to the hypothesis) start at line 137

145-146: "for six of the eight"? This is not correct as DC(B) was not significant; the next sentence says it correctly, for 2 only; in fact, for 2 of the 12 with sufficient data.

Table 2: if some of the data is unreliable (line 649), perhaps it should be omitted or at least indicated more clearly in the table (not only in the caption)

151: add "Table 1" at the end of the bracket (I guess Table 1 provides the sample size)

152: avoid explaining patterns that are not significant.

Fig. 3: This doesn't seem the correct type of plot for this data. Boxplots would be more appropriate (they can also include data points). The font is very small (e.g., species names). For

the fitted lines, did you merge the different species?!

172: significantly less? Are precipitation variables (Fig. 4) significantly different from 0? So this paragraph (lines 170-177) need revision. Fig. 4 suggests that there is only one significant climate shift (temperature), and not three as mentioned in the text, so the next paragraph (178-...) and the Discussion also need revision.

182-183: Unclear. Following table 2, leading and source populations of *Quercus chrysolepis*, *Abies grandis* and *P. tremuloides* (in burned sites) are not significantly different, so fire is not moving them. Significantly faster (line 183, 187, 188 ...)?

241: serotinous species (or varieties) should be excluded from the analysis

428-429: this suggest that to support the hypothesis you need both tests to be significant; this seems appropriate to reduce false positives. So your hypothesis is only supported for 2 (on the 12 species with data), and even for one of the species the sample size is small (L.d.).

Review of revision #2 of Avery and Field, “Wildfire and the Rate of Tree Range Shifts in the American West”

Philip Higuera, University of Montana, 10 May, 2021.

The text is improved again with this second revision, and many of the reviewer comments have been well addressed. I suspect it sounds like I am repeating much of what I wrote in previous reviews, and this may be frustrating to the authors. I believe this is true: I find myself commenting on many of the same core issues raised by myself and others in review 1 and 2.

[1] The explanation for the geographic extent of the current study – in the response to reviews, and in the text (L93-97; and L 346-350) -- is not acceptable in my opinion, IF “the American West” is retained in the title. The study simply does not cover “the American West” -- e.g., Fig. 1 is a subset of the western US. That WY was left out unintentionally is not a strong rationale for a final paper. And, essentially dismissing forests in UT, CO, NM (and AZ) as “disjunct patches” (L 95) is really hard to swallow. These forests are not “disjunct patches”.... There are major programs in all of these regions focused on forests (e.g., USDA research labs, entire academic research careers, timber and tourism economies). It’s fine that WY is “only” 10% of the FIA data, by number, but thus, plus the other states this leaves out a significant portion of the American West (in geographic and climate space). I don’t think a paper can have it both ways: either sample what the title reflects, or if the FIA data don’t fit the desired sampling criteria, then reflect this in the title. This, from Dobrowski et al. (2015 – Title = “Forest structure...in western US tree species”), seems like a much more defensible and reasonable description of FIA data to study western forests: “The FIA plots used in this analysis came from the 11 western US states and were inventoried between 1984 and 2011 with sampling intensity varying among states and between the analyses we present (demographic niche differences versus recruitment models).”

[2] While much of the singular focus on competition has been removed from the paper, to its benefit, there is still some text that remains confusing or misleading. E.g., L 68-71: As noted before, I don’t think any single pattern could “definitively prove” anything, and that’s not a realistic expectation for any single study; and, the “absence of an effect of...would provide strong grounds for rejecting the hypothesis” statement is complicated, and it’s not clear it’s correct. As noted before, couldn’t one find evidence for fire-affected range shifts, even if competition dynamics were unchanged? The key missing mechanism here (and specifically in L 73) is that killing adult trees. Killing adult trees, which could survive longer in a range currently unsuitable for seedlings of the same species...does not require one to invoke competition as an explanation, and yet it would accelerate range shifts.

[3] L 176: Here and throughout – again a comment made in review 1-2 – the verb “shift” implies change over time. The statistics and patterns document differences in space. The inference is a shift in time. Describing the results as if they reflect change over time is misleading.

Detailed comments, focused on new text:

L 80: “initial pre-fire...” – why “initial”? Fires are happening throughout existence of these ecosystems – this suggests fire as an extrinsic factor in these ecosystems, which it is not. Return to “pre-fire conditions” would be fit better. Here, and L 84.

L 148: The “...not statistically significant” followed by “ $p < 0.05$ ” is confusing, and may be to other readers too – is the sign switched, or does this deserve a brief description of the null hypothesis?

L 171: Should be “historical climatic niche” – not “historic” (unless it really is historic).

L 178: Text should be clear here that this is speculation, reg. migration? Again, just reading this, is sounds like the study measured migration; it did not. Since this is the results section – the data should be described without inference.

L 266: Suggest this new edit here not use “hillside” – as it’s not relevant. The topography is not the issue here, and stand-replacing fire also occurs on flat ground. “A gap in the canopy or large high-severity patch potentially promotes establishment, if seed is available, through reduced competition.”

Figure 2: Title of this should be “range differences” – it’s not range shifts. No shift has been measured. This contributes to the paper feeling misleading.

Figure 3: If the climate data are expressed as z-scores, I’d expect the mean to 0 and standard deviation to be 1 (as in Fig. 4). Why is 0 always at the far lower end, or below the limits of the y-axis? It seems more like the mean is near 1, in which case the values were just divided by the mean? Does (I suspect) this reflect the statistical model, and the y-axis is relative influence (or something like that)? If so, the caption text should just clarify this a bit, as it reads as if we should be able to see the z-score on the y-axis.

Line by Line Response

Reviewer #2:

The text is improved again with this second revision, and many of the reviewer comments have been well addressed. I suspect it sounds like I am repeating much of what I wrote in previous reviews, and this may be frustrating to the authors. I believe this is true: I find myself commenting on many of the same core issues raised by myself and others in review 1 and 2.

[1] The explanation for the geographic extent of the current study – in the response to reviews, and in the text (L93-97; and L 346-350) -- is not acceptable in my opinion, IF “the American West” is retained in the title. The study simply does not cover “the American West” -- e.g., Fig. 1 is a subset of the western US. That WY was left out unintentionally is not a strong rationale for a final paper. And, essentially dismissing forests in UT, CO, NM (and AZ) as “disjunct patches” (L 95) is really hard to swallow. These forests are not “disjunct patches”.... There are major programs in all of these regions focused on forests (e.g., USDA research labs, entire academic research careers, timber and tourism economies). It’s fine that WY is “only” 10% of the FIA data, by number, but thus, plus the other states this leaves out a significant portion of the American West (in geographic and climate space). I don’t think a paper can have it both ways: either sample what the title reflects, or if the FIA data don’t fit the desired sampling criteria, then reflect this in the title. This, from Dobrowski et al. (2015 – Title = “Forest structure...in western US tree species”), seems like a much more defensible and reasonable description of FIA data to study western forests: “The FIA plots used in this analysis came from the 11 western US states and were inventoried between 1984 and 2011 with sampling intensity varying among states and between the analyses we present (demographic niche differences versus recruitment models).”

In the interest of thoroughness, we decided to re-analyze our data using all available FIA data from Northwestern Forested Mountains and Marine West Coast Forest ecoregions in the continental US. This added FIA data from UT, CO, NM, and WY. Even so, this area does not include all western US states so we’ve changed the title to the more specific *Wildfire, Climate, and the Range of Trees in Forested Ecoregions of the Western US.*

[2] While much of the singular focus on competition has been removed from the paper, to its benefit, there is still some text that remains confusing or misleading. E.g., L 68-71: As noted before, I don’t think any single pattern could “definitively prove” anything, and that’s not a realistic expectation for any single study; and, the “absence of an effect of...would provide strong grounds for rejecting the hypothesis” statement is complicated, and it’s not clear it’s correct. As noted before, couldn’t one find evidence for fire-affected range shifts, even if competition dynamics were unchanged? The key missing mechanism here (and specifically in L 73) is that killing adult trees. Killing adult trees, which could survive longer in a range currently unsuitable for seedlings of the same species...does not require one to invoke competition as an explanation, and yet it would accelerate range shifts.

We’ve reworked Lines 68-71 (now Lines 69-71) and have removed any language related to ‘proving’ this hypothesis.

“Wildfire, because it reduces vegetation cover and therefore reduces some aspects of plant-plant competition, opens a window on the hypothesis that competitors can accelerate climate-related range expansion.”

In principle, the systematic death of mature trees in specific parts of the habitat could lead to a niche difference between plots with seedlings only and trees plus seedlings. But these

would make it past the screens only if the direction of the difference was the same in burned and unburned plots. In addition, it seems unlikely that mature trees would be dying specifically in the part of the habitat space that tends to keep populations in their historic climatic niches. Thank you for identifying this possibility, which we discuss on lines 210-212.

[3] L 176: Here and throughout – again a comment made in review 1-2 – the verb “shift” implies change over time. The statistics and patterns document differences in space. The inference is a shift in time. Describing the results as if they reflect change over time is misleading.

We’ve carefully worked through the manuscript and have replaced everything that implies that we measured change over time with alternative language, particularly in the results section. ‘Range shift’ has been replaced throughout the text with “seedling-only range displacement”. Our reporting of the results is careful to not imply measurements over time. For example (Lines 158-160) now read

“The three subalpine species, which do not exhibit lower MTWM or higher MSP in seedling-only plots, are similar to each other in that they all have *higher* MTWM and lower MTCM in seedling-only plots in both burned and unburned plots”

Also, in the interest of being more concise and accurate with our language we’ve changed “potential leading-edge” and “source population” to “seedling-only” “tree-plus-seedling plots” as evidenced above and throughout the text.

Detailed comments, focused on new text:

L 80: “initial pre-fire...” – why “initial”? Fires are happening throughout existence of these ecosystems – this suggests fire as an extrinsic factor in these ecosystems, which it is not. Return to “pre-fire conditions” would be fit better. Here, and L 84.

We’ve removed the term ‘initial’ from Line 81, and have changed lines 84-85 to “...the ability of an ecosystem to return to the pre-existing species composition after disturbance”

L 148: The “...not statistically significant” followed by “ $p < 0.05$ ” is confusing, and may be to other readers too – is the sign switched, or does this deserve a brief description of the null hypothesis?

With the comprehensive reanalysis of the data, this sentence no longer exists

L 171: Should be “historical climatic niche” – not “historic” (unless it really is historic).

The sentence looks much different now, but we’ve replaced “historic” with “historical” (now Line 176)

L 178: Text should be clear here that this is speculation, reg. migration? Again, just reading this, it sounds like the study measured migration; it did not. Since this is the results section – the data should be described without inference.

This section has been entirely re-written with our new results. In light of this and an earlier comment of yours, we were careful to remove language implying range shifts, migration, and temporal shifts. Instead, the language more accurately reflects the analysis. E.g. “MSP decreased across the study area and five species have seedling-only plots with significantly higher values of MSP than tree-plus-seedling plots” (Lines 169-172),

L 266: Suggest this new edit here not use “hillside” – as it’s not relevant. The topography is not the issue here, and stand-replacing fire also occurs on flat ground. “A gap in the canopy or large high-severity patch potentially promotes establishment, if seed is available, through reduced competition.”

As suggested this sentence now reads “A gap in the canopy or large high-severity burn patch potentially promotes establishment through reduced competition” (Lines 265-266)

Figure 2: Title of this should be “range differences” – it’s not range shifts. No shift has been measured. This contributes to the paper feeling misleading.

Here, as elsewhere in the text, we’ve replaced the term ‘range shift’ with ‘range displacement’ or ‘seedling-only range displacement’ to increase the precision of the language.

Figure 3: If the climate data are expressed as z-scores, I’d expect the mean to 0 and standard deviation to be 1 (as in Fig. 4). Why is 0 always at the far lower end, or below the limits of the y-axis? It seems more like the mean is near 1, in which case the values were just divided by the mean? Does (I suspect) this reflect the statistical model, and the y-axis is relative influence (or something like that)? If so, the caption text should just clarify this a bit, as it reads as if we should be able to see the z-score on the y-axis.

When scaling these values in R we used the scale() function where “center = FALSE”, standardizing values by dividing by the root mean square. We’ve updated the Figure 3 caption to clarify.

Reviewer #3:

The manuscript tests a nice idea that is worth persuading. The results show a clear pattern supporting the hypothesis at least for one species (of the 12 species with data); another species also show the pattern but the authors recognise the small sample size; three additional species partially show the pattern, one of them include serotinous populations and thus their range shift may be inflated (line 241); and some species seems to shift in the opposite direction, i.e., to warmer sites (line 293-...). So I feel that there is an interesting pattern although I must recognise that the pattern is not very strong.

The writing still needs improving, it should be written with more precision; the quality of the figures is also limited.

Specific comments by line number

20: For the abstract, the number of species considered is more important than the number of plots.

We’ve added the initial number of tree species sourced to this sentence of the abstract (Line 21)

45: “(1) the environment or disturbance” (typically disturbance is not considered an environmental variable)

We’ve replaced ‘increased disturbance’ with another climatic variable, ‘precipitation seasonality’ (Line 49)

71: The absence of effect cannot reject an (alternative) hypothesis!

We've softened the language of this, and have removed mention of rejecting this hypothesis, which is not the explicit hypothesis we set out to test. This sentence now reads "Wildfire, because it reduces vegetation cover and therefore reduces some aspects of plant-plant competition, opens a window on the hypothesis that removing competitors can accelerate climate-related range expansion." (Lines 69-71)

84-86: sentence unclear

This sentence now reads "The impact of fire on species regeneration (e.g. by altering seedbank composition) was cited as an important driver of the observed state shifts" (Line 85-87)

86: "south" – are you still talking about Alaska??

We've removed the geographic contextualization altogether because it was more confusing than constructive.

Line 87 now reads "Fire and climate change, in concert, can affect tree regeneration"

Fig. 1: it is not easy to differentiate the size of the symbols (number of plots) in the map.

We've readjusted sizing and have added color to help differentiate the plot density

118-119: excluded? Because it doesn't support your hypothesis? Perhaps the species that do not behave as hypothesised (4 of 12) should not be removed from Table 1, they are part of the results.

We revised the description of the species screen and the species cut by it to clarify the motivation and implications. The analysis doesn't make sense if the direction of the range displacement in the seedling-only plots is not in the same direction as in the sapling plots. Similarly, it doesn't make sense if the direction of the range displacement in seedling-only plots is different between burned and unburned plots. These screens are our primary tool for minimizing the risk that the observed pattern is contaminated by seedlings growing in places where they will not reproduce or fire adaptations that are independent of climate-related range displacements. These screens are now discussed in lines 112-117 and 193-212. Figure A6 in the supplementary material illustrates why the analysis loses meaning when the species are not filtered by directional inconsistency in burned plots, *Lithocarpus densiflorus* (otherwise excluded) shows evidence of seedling only range displacement in burned and unburned plots, but because the displacements are in different directions, in climate space, it is unlikely that both are a response to climate.

Also in Table 1 should mark the species that are resprouters or serotinous, e.g., with an (R) or (S) after the name. And mention that the number of seedlings of these species are likely to be overestimated.

We decided to add this information to Table 2 instead of Table 1, because serotiny and resprouting are relevant to the seedling-only range displacement metrics which are described in Table 2. We've appended R and S to species names and in the caption pointed the reader to further discussion in the main text.

124, 125: repetition (0.82-0.56 vs less than 0.82)

Removed the 2nd reporting of values (Lines 126-128)

113-136: The first section of the results (“Evidence of range shifts in unburned plots”) doesn’t seem to respond to your hypothesis; removing it doesn’t seem to affect the paper. Results (in relation to the hypothesis) start at line 137

The paper’s core goal is to test whether the range displacement in seedling-only plots is greater in burned than in unburned plots, for species in which the displacement of seedlings or saplings from tree-plus seedling plots is in the same direction for burned and unburned plots. Unless there is evidence for seedling-only range displacement in unburned plots, the test doesn’t have a starting point. In essence, we are using the unburned plots to define the direction, in climate space, of the climatically driven range displacement. We’ve expanded the explanation of the importance of confirming potential range shifts in unburned plots in the results (Lines 120-121)

We’ve also tweaked the description of the hypothesis to clarify that this paper’s focus is the relationship between fire and range shifts for species that are already responding to climate change. The hypothesis in Lines 93-94 now reads “forest fires can facilitate range expansions in tree species responding to contemporary climate change”

145-146: “for six of the eight”? This is not correct as DC(B) was not significant; the next sentence says it correctly, for 2 only; in fact, for 2 of the 12 with sufficient data.

This text has been altered to emphasize only the species results with statistical significance. “We found that CD_B is greater than CD_U for both *Pseudotsuga menziesii* and *Quercus chrysolepis* ($p < .05$)” (Lines 144-145)

Table 2: if some of the data is unreliable (line 649), perhaps it should be omitted or at least indicated more clearly in the table (not only in the caption)

We dropped the use of ‘unreliable’ and replaced it with 95% and 90% confidence intervals around the $CD_B - CD_U$ metric, now reported in Table 2.

151: add “Table 1” at the end of the bracketed (I guess Table 1 provides the sample size)

We removed this sentence with an unrelated change in results

152: avoid explaining patterns that are not significant.

Here and elsewhere we’ve removed the reporting and discussion of much of the results that are statistically insignificant (Lines 120-179). In the places where we report trends across species with $p < 0.05$ and $p > 0.05$, we state this explicitly (Line 147)

Fig. 3: This doesn’t seem the correct type of plot for this data. Boxplots would be more appropriate (they can also include data points). The font is very small (e.g., species names). For the fitted lines, did you merge the different species?!

We’ve remade Figure 3 in the interest of clarity. As suggested, we now use paired boxplots to exhibit the differences between the magnitudes of climatic niche differences in burned and unburned plots. We merged the climate values of different species that exhibited

climatic differences in the same direction in order to increase the sample size and robustness of the regression.

Figure A8 in the supplementary material showcases each climate variable and directional difference (as Figure 3 did in the previous draft), and we pared Figure 3 down to show only the 2 climate variables where wildfire occurrence was correlated with greater seedling-only range displacement.

172: significantly less? Are precipitation variables (Fig. 4) significantly different from 0? So this paragraph (lines 170-177) need revision. Fig. 4 suggests that there is only one significant climate shift (temperature), and not three as mentioned in the text, so the next paragraph (178-...) and the Discussion also need revision.

All four variables exhibit statistically significant shifts. We've added means and 95% confidence intervals to Fig. 4 to make this more clear.

182-183: Unclear. Following table 2, leading and source populations of *Quercus chrysolepis*, *Abies grandis* and *P. tremuloides* (in burned sites) are not significantly different, so fire is not moving them. Significantly faster (line 183, 187, 188 ...)?

This section has been re-written to incorporate the new results, including altered species groupings. In this revised section we discuss only climatic differences that are statistically significant (Lines 174-180)

241: serotinous species (or varieties) should be excluded from the analysis

The revised text has increased discussion of which species are serotinous (*P. contorta* var. *latifolia*) in Table 2 and in the text (Lines 236-245). Based in the new analysis *Pinus contorta* is no longer included in additional analyses in which results are averaged across species (e.g. Figure 3).

428-429: this suggest that to support the hypothesis you need both tests to be significant; this seems appropriate to reduce false positives. So your hypothesis is only supported for 2 (on the 12 species with data), and even for one of the species the sample size is small (L.d.).

In response to reviewer comments, we've developed a metric that explicitly tests for differences in seedling-only range displacement between burned and unburned plots, based on: the bootstrapped differences between the centroid distances ($CD_B - CD_U$).

The species where ($CD_B - CD_U$) is significant also meet the requirements that we described in our previous drafts for evidence of fire-facilitated potential range shifts: Schoener's Ds of burned and unburned species are both statistically significant, the burned Schoener's D is less than unburned Schoener's D, the Centroid Distances are both statistically significant, and burned centroid distance is greater than unburned centroid distance.

**The updated methods section now describes ($CD_B - CD_U$) as the primary metric for comparing the seedling only range displacement between burned and unburned plots
From Lines 429 – 433:**

“we use the difference between CD_B (centroid distance, burned) and CD_U (centroid distance, unburned) as our primary measure of whether the magnitude of SORDs are different between burned and unburned plots. We bootstrap the calculation of $CD_B - CD_U$ using 20,000 replicates to produce 90% and 95% confidence intervals.”

Reviewer comments, fourth round review:

Reviewer #1 (Remarks to the Author):

Review for: Wildfire, climate, and the range of trees in forested ecoregions of the western US
I am impressed with the efforts made by the authors to address the previous methodological and interpretive issues raised by several reviewers. I do believe the concerns have been addressed with the more conservative SORD approach adopted by the authors in this iteration of the paper. I also feel the description of their rather complicated approach is nicely written and relatively easy to digest. I don't know if my repeated exposure to the paper makes it so – but regardless, I do feel the paper is ready for publication. I do want to state that I must trust that the authors' have properly executed their analyses, as they are outside the scope of analyses I have done. But they do seem well executed and transparently explained/interpreted. A few minor comments below:
Title: I suggest saying "distribution range" to make clear what type of range is being considered. Or, maybe just substitute in "distribution" for "range".
Line 169: Do the authors really mean bias here? Or, rather, "trend" or similar?
Line 176: Again, do the authors mean bias? Or is this a directional finding?
Line 289, 293 and anywhere else for the remainder of the manuscript: Bias? Or trend?

Reviewer #3 (Remarks to the Author):

It would be important that the authors provide all data and details so somebody can easily repeat and reanalysis the dataset.

Specific comments by line number

21: The work presented doesn't analyse 110 tree species, only 8 species (Table 1). 110 is the number of species in the dataset considered, but most species were discarded for the analysis.
69-73: Text needs to be improved. "opens a window of the hypothesis" is confusing. Please, make clear your hypothesis. What about removing this sentence? Do you really need to talk about competition here?
110-118 (Species selection): I would say that this section is not a result, it may need to be moved to methods.
167-173: where are the results related to this text?
308-310: landscape fragmentation is also a problem when using climatic distances as a proxy for spatial distances.
392: I think the acronym SORD has not yet been explained.
434: "Additional analyses" Could you be a bit more specific with this heading?
453: unclear ("consistently" vs "for selected species")
Table 1: It is hard to understand values like "97% ± 33" when the maximum agreement value is 100% (all); something must be wrong there.
Table 2. The two significant species are different from the previous version! Does this mean that the results are very sensitive to the method? Is your method robust now? In any case, please consider $p < 0.05$ as significant, not $p < 0.1$.

REVIEWERS' COMMENTS

Reviewer #1 (Remarks to the Author):

Review for: Wildfire, climate, and the range of trees in forested ecoregions of the western US
I am impressed with the efforts made by the authors to address the previous methodological and interpretive issues raised by several reviewers. I do believe the concerns have been addressed with the more conservative SORD approach adopted by the authors in this iteration of the paper. I also feel the description of their rather complicated approach is nicely written and relatively easy to digest. I don't know if my repeated exposure to the paper makes it so – but regardless, I do feel the paper is ready for publication. I do want to state that I must trust that the authors' have properly executed their analyses, as they are outside the scope of analyses I have done. But they do seem well executed and transparently explained/interpreted. A few minor comments below:

Title: I suggest saying “distribution range” to make clear what type of range is being considered. Or, maybe just substitute in “distribution” for “range”.

We changed the title to one recommended by the editorial staff: *Forest fires and climate-induced tree range shifts in the western US*

Line 169: Do the authors really mean bias here? Or, rather, “trend” or similar?

Line 176: Again, do the authors mean bias? Or is this a directional finding?

Line 289, 293 and anywhere else for the remainder of the manuscript: Bias? Or trend?

We replaced ‘bias’ with ‘trend’ wherever the word occurred in reference to SORD direction. (Line 185, 192-193, 317, 321, Figure 4 caption)

Reviewer #3 (Remarks to the Author):

It would be important that the authors provide all data and details so somebody can easily repeat and reanalysis the dataset.

Specific comments by line number

21: The work presented doesn't analyse 110 tree species, only 8 species (Table 1). 110 is the number of species in the dataset considered, but most species were discarded for the analysis.

We removed reference to the 110 tree species in this clause so that it now focuses solely on the number and distribution of plots in the study area. The sentence that follows already refers to the 8 tree species analyzed.

Lines (17-22) now read “In this study we test the sensitivity of tree range shifts (measured as the difference between seedling and mature tree ranges in climate space) to wildfire occurrence, using 74,069 Forest Inventory Analysis plots across nine states in the western United States. Wildfire significantly increased the seedling-only range displacement for 2 of the 8 tree species in which seedling-only plots were displaced from tree-plus-seedling plots in the same direction with and without recent fire”

69-73: Text needs to be improved. “opens a window of the hypothesis” is confusing. Please,

make clear your hypothesis. What about removing this sentence? Do you really need to talk about competition here?

We feel that the reference to competition is an important feature of our work, so instead of removing the sentence we edited it to improve its clarity:

“Wildfire, because it reduces vegetation cover and therefore reduces some aspects of plant-plant competition, provides an entry point for exploring the hypothesis that removing competitors can accelerate climate-related range expansion” (Lines 77-79)

110-118 (Species selection): I would say that this section is not a result, it may need to be moved to methods.

167-173: where are the results related to this text?

The data to support this text is shown in what is now Supplementary Figure 2. We added reference to this at the end of the section (Line 189)

308-310: landscape fragmentation is also a problem when using climatic distances as a proxy for spatial distances.

We added reference to habitat fragmentation as an additional landscape feature that complicates that the relationship between climatic and spatial distance.

Lines 338-340 now read:

“Some uncertainty comes from using climatic distance as a proxy for spatial distance (Equation 1). The relationship is certainly not fixed, particularly across topographically diverse regions or fragmented habitats.”

392: I think the acronym SORD has not yet been explained.

We added the definition for SORD to the first instance of the term in the methods section (Lines 427-428)

434: “Additional analyses” Could you be a bit more specific with this heading?

A more specific heading wouldn’t work here because we describe a diversity of actions taken. Instead, we broke the “Additional Analyses” into two sections, “Continued SORD Analyses” and “Evaluation of Methodological Robustness.” (Lines 472-509)

453: unclear (“consistently” vs “for selected species”)

We replaced “select species” with “a specific set of species” to clarify that the same set of species showed consistent SORDs across different sets of climatic variables. The clause now reads:

“Schoener’s D and Euclidean centroid distance consistently indicated SORDs were greater in plots that burned for a specific set of species (Supplementary Tables 3.1-3.3).” (Lines 493 – 495)

Table 1: It is hard to understand values like “97% ± 33” when the maximum agreement value is 100% (all); something must be wrong there.

Our understanding is that confidence intervals can exceed the limits of a parameter and that values such as “97% ± 33” are best interpreted as “64% - 100%” (source: Prof. Philip)

Stark's online statistics textbook). To increase clarity, we presented the data as a range instead, e.g. "64% - 100%"

Table 2. The two significant species are different from the previous version! Does this mean that the results are very sensitive to the method? Is your method robust now? In any case, please consider $p < 0.05$ as significant, not $p < 0.1$.

While the results are likely to have some sensitivity to the methodological changes implemented since the last draft (e.g. the use of confidence intervals in species vetting and a more sophisticated calculation of significant differences in SORDs between burned and unburned sites), it's likely that much of the difference in results can be attributed to the increase in study area.